# GraSP: Simple yet Effective Graph Similarity Predictions

## Abstract

Graph similarity computation (GSC) is considered one of the essential operations because of its wide range of applications in various fields. Graph Edit Distance (GED) and Maximum Common Subgraph (MCS) are the most popular graph similarity metrics. However, calculating exact GED and MCS is a complex task that falls under the category of NP-hard problems. Consequently, state-of-the-art methodologies learn data-driven models leveraging graph neural networks (GNNs) for estimating GED and MCS values. A perceived limitation of these approaches includes reliance on computationally expensive cross-graph node-level interaction components but to little avail. Instead of building up complicated components, we aim to make the complicated simple and present GraSP, a simple yet highly effective approach for GSC. In particular, to achieve higher expressiveness, we design techniques to enhance node features via positional encoding, employ a graph neural network backbone with a gating mechanism and residual connections, and develop a multi-scale pooling technique to generate meaningful representations. We theoretically prove that our method is more expressive and passes 1-WL test performance capabilities. Notably, GraSP is versatile in accurately predicting GED and MCS metrics. In extensive experiments against numerous competitors on real-world datasets, we demonstrate the superiority of GraSP over prior arts regarding effectiveness and efficiency. The source code is available at `https://anonymous.4open.science/r/GraSP`.

## 1 Introduction

A graph is a data structure to model relationships between objects. Graphs are ubiquitous in the real world and have critical applications in bioinformatics, chemoinformatics, and social networks. Calculating the similarity between graphs is a crucial operation with a range of subsequent applications, such as ranking related documents in information retrieval (Lee et al., 2008), disease prediction in bioinformatics (Borgwardt & Kriegel, 2007), and code similarity detection (Li et al., 2019). Graph Edit Distance (GED) (Bunke & Allermann, 1983) and Maximum Common Subgraph (MCS) (Bunke & Shearer, 1998) are two popular graph similarity/distance metrics because they are domain-agnostic. However, the computation of both GED and MCS is NP-hard (Zeng et al., 2009; Liu et al., 2020), and in recent studies, computing the exact GED between graphs with more than sixteen nodes is infeasible (Blumenthal & Gamper, 2020).

Methods for exact GED computation have mainly used the filter-verification framework with exponential complexity, and recent work includes Kim et al. (2019); Kim (2020); Chang et al. (2020). In order to reduce the computation cost, some traditional combinatorial approximation methods such as Beam (Neuhaus et al., 2006), Hungarian (Riesen & Bunke, 2009) and VJ (Fankhauser et al., 2011) have been proposed at a cost of accuracy drop. However, these methods have the complexity of sub-exponential or cubic. In recent years, thanks to the powerful expressiveness of graph neural networks (Bai et al., 2019; Li et al., 2019; Bai et al., 2020; Ling et al., 2023; Qin et al., 2021; Zhuo & Tan, 2022; Ranjan et al., 2022) have been proposed. These methods have three main components, namely the node and graph embedding module, the cross-graph node-level interaction module, and the similarity computation module. Most of these works use the cross-graph node-level interactions, either explicitly or implicitly. However, this module typically takes quadratic term time during offline training and online inference, and is expensive.

In this paper, we present GRASP, a method that is simple but effective to grasp the intrinsic representations for GRAph Similarity Prediction. Remarkably, with careful designs, GRASP can achieve superior performance for effective graph similarity prediction and is efficient for training and inference. Specifically, we design a technique to enhance node features via positional encoding to consider the global topological patterns in a graph, adopt a GNN backbone with a gating mechanism and residual connections to generate meaningful representations, and concatenate the representations at each convolutional layer to retain information about the neighbors at each hop. Moreover, we devise a multi-scale pooling technique to combine the merits of pooling techniques for effectiveness. Compared with conventional GNN layers, like graph convolutional network (GCN) (Kipf & Welling, 2017) or graph isomorphism network (GIN) (Xu et al., 2019) that are with restricted expressive power by first-order Weisfeiler-Leman (1-WL) graph isomorphism test (Weisfeiler & Leman, 1968), we theoretically prove that GRASP can indeed achieve higher expressiveness by passing 1-WL test, which potentially benefits the practical effectiveness to predict graph similarity values accurately. We have conducted extensive experiments to compare with 8 competitors on 4 datasets under various settings. Our method GRASP consistently achieves the best performance under almost all metrics on all datasets for accurately estimating both GED and MCS. We also conduct ablation study, efficiency evaluation, case study, etc, to further validate the power of GRASP.

In summary, our contributions are as follows:

• We propose a novel graph similarity prediction approach, GRASP that achieves superior performance via a series of technical designs that are rational and effective.

• For the problem of graph similarity prediction, to our knowledge, we are the first to use positional encoding to improve the performance. We also design a multi-scale technique on an advanced GNN backbone to enhance our method.

• We conducted comprehensive experiments to validate that our method GRASP achieves superior performance on estimating two metrics, GED and MCS, compared with existing state-of-the-art.

## 2 PROBLEM STATEMENT

A graph $\mathcal{G}$ is a data structure consisting of a set of nodes $\mathcal{V}$ and a set of edges $\mathcal{E}$, i.e., $\mathcal{G} = (\mathcal{V}, \mathcal{E})$. The number of nodes and edges are $|\mathcal{V}|$ and $|\mathcal{E}|$, respectively. Given a graph database $\mathcal{D}$ containing a collection of graphs, we aim to build an end-to-end model to accurately estimate the similarity values of graph pairs. The model is designed to predict multiple similarity/distance metrics, including Graph Edit Distance (GED) and Maximum Common Subgraph (MCS). GED is the edit distance between two graphs $\mathcal{G}_1$ and $\mathcal{G}_2$, i.e., the minimum number of edit operations to convert $\mathcal{G}_1$ to $\mathcal{G}_2$, where edit operations include adding/removing a node, adding/removing an edge, and relabeling a node. MCS refers to find the largest subgraph that is common to the two graphs $\mathcal{G}_1$ and $\mathcal{G}_2$. Following the choice of (Bai et al., 2020), we require the MCS to be connected. Figure 1 shows GED and MCS examples. Let $m$ be the number of labels, and we can define the node feature matrix $\mathbf{X} \in \mathbb{R}^{|\mathcal{V}| \times m}$ of a graph $\mathcal{G}$. During the training phase, a training sample consisting of a set of graph pair $(\mathcal{G}_1, \mathcal{G}_2) \in \mathcal{D} \times \mathcal{D}$ with ground-truth similarity value $s(\mathcal{G}_1, \mathcal{G}_2)$. In the inference phase, the model predicts the similarity/distance of unseen graph pairs.

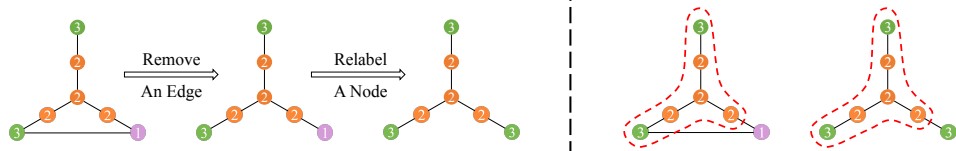

Figure 1: GED and MCS examples from AIDS700nef dataset. Left: GED is 2 and right: MCS is 6.

## 3 OUR PROPOSED APPROACH: GRASP

In this section, we present the proposed method GRASP. The architecture of GRASP is presented in Figure 2. As shown in Figure 2, in the node feature pre-processing, we first enhance node features by concatenating positional encoding of the node. This enhancement considers global graph topological features. Second, the node embedding module preserves the enhanced node features via a RGGC backbone into node embeddings. Third, in the graph embedding module, we devise a

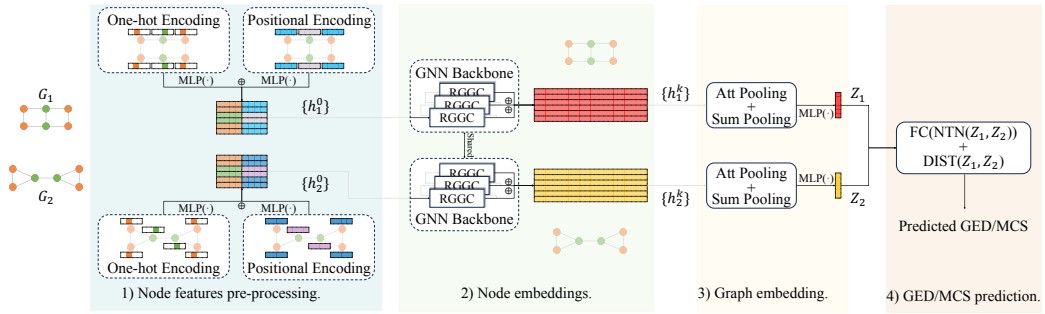

Figure 2: The architecture of GRASP.

multi-scale node embedding pooling technique to obtain graph embeddings. Lastly, the prediction module predicts the similarity/distance between the two graph embeddings.

## 3.1 ENHANCED NODE FEATURES VIA POSITIONAL ENCODING

Previous works like Bai et al. (2019; 2020); Ranjan et al. (2022) obtain features using the one-hot encoding of node labels. Specifically, every node $i$ has $\mathbf{x}_i \in \mathbb{R}^m$ to represent its node label, where $m$ is the number of labels in the graph database. Then $\mathbf{x}_i$ is transformed by an MLP to $\boldsymbol{\mu}_i \in \mathbb{R}^d$ (Ranjan et al., 2022), which is then used as the input of GNN layers.

However, only using node labels as node features to be the input of Message-passing graph neural networks (MPGNNs) cannot pass the first-order Weisfeiler-Leman (1-WL) graph isomorphism test (Weisfeiler & Leman, 1968). Specifically, MPGNNs are a series of graph neural networks that update node representations based on message passing and by aggregating neighbor information, a node $i$ in a graph can be represented as:

$$\mathbf{h}_i^{(\ell)} = f\left(\mathbf{h}_i^{(\ell-1)}, \left\{\mathbf{h}_j^{(\ell-1)}\middle| j \in \mathcal{N}(i)\right\}\right), \tag{1}$$

where $\ell$ is l-th layer of the stack of the MPGNN layers, $\mathcal{N}(i)$ denotes the neighbors of node $i$.

It has been shown by Xu et al. (2019) that MPGNNs can perform up to, but not beyond, the level of the 1-WL test (Weisfeiler & Leman, 1968). In general, the 1-WL test can effectively distinguish non-isomorphic graphs. However, since MPGNNs only aggregate information from their direct neighbors, they fail to capture higher-order structural properties and the global topology of the graph, thus limiting the expresive capability of MPGNNs. We take a case that MPGNNs fail to distinguish (Sato, 2020), as shown in Figure 3. MPGNNs produce the same set of node embeddings for these two non-isomorphic graphs. Thus, the same set of node embeddings causes our model to give incorrect predictions that the GED is 0, which apparently should not be 0.

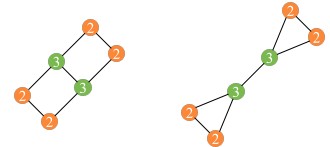

Figure 3: MPGNNs fail to differentiate non-isomorphic graphs.

To mitigate this issue and enhance the expressiveness of our method, we propose to enhance node features with positional encoding that can exceed the expressive capacity of the 1-WL test. Specifically, we use the random walk method for position encoding, RWPE, which has been empirically proven to work by Dwivedi et al. (2022). Unlike the distance encoding method proposed by Li et al. (2020) that uses random walks to learn the distance of all pairs of nodes in the graph, RWPE uses only the probability of a node landing on itself, i.e., the diagonal part of the random walk matrix. Given the random walk length $k$, we can use RWPE to precompute the positional features $\mathbf{p}_{i_{\text{init}}} \in \mathbb{R}^k$ of a node $i$ in the graph, denoted as $\mathbf{p}_{i_{\text{init}}} = \left\{\mathbf{RW}_{ii}^{\ell}\middle| \ell = 1, 2, ..., k\right\}$, where $\mathbf{RW} = \mathbf{A}\mathbf{D}^{-1}$ is the random walk matrix obtained by adjacency matrix $\mathbf{A}$ and diagonal degree matrix $\mathbf{D}$ of a graph. Then we also transform the RWPE $\mathbf{p}_{i_{\text{init}}}$ into $\mathbf{p}_i \in \mathbb{R}^d$ by an MLP.

We concatenate the transformed positional encoding $\mathbf{p}_i$ and the encoding $\boldsymbol{\mu}_i$ both in $\mathbb{R}^d$ of node $i$ together to get its enhanced representation $\mathbf{h}_i^{(0)} \in \mathbb{R}^{2d}$, which will be the input of GNN backbone,

$$\mathbf{h}_i^{(0)} = \text{CONCAT}\left(\boldsymbol{\mu}_i, \mathbf{p}_i\right). \tag{2}$$

## 3.2 Multi-Scale Pooling on RGGC

The enhanced node features $\mathbf{h}_i^{(0)}$ obtained in Section 3.1 are then fed into a graph neural network consisting of $n$ layers of ResGatedGraphConv (RGGC layers) (Bresson & Laurent, 2017) to learn the hidden representations of the nodes. The RGGC layers can leverage a gating mechanism where gating units learn to control information flow through the network. Moreover, residual connections are used in RGGC layers to help with gradient flow during training. With these two techniques incorporated, the RGGC layers can learn complex patterns in graph data, and therefore, are more powerful and versatile than basic graph convolutional layers like GCNs and GINs. At the $\ell$-th layer, the node representation $\mathbf{h}_i^{(\ell)}$ is

$$\mathbf{h}_i^{(\ell)} = \mathbf{h}_i^{(\ell-1)} + \text{ReLU}\left(\mathbf{W}_1 \mathbf{h}_i^{(\ell-1)} + \sum_{j \in \mathcal{N}(i)} \eta_{i,j} \odot \mathbf{W}_2 \mathbf{h}_j^{(\ell-1)}\right), \tag{3}$$

where $\mathbf{W}_1 \in \mathbb{R}^{2d \times 2d}$ and $\mathbf{W}_2 \in \mathbb{R}^{2d \times 2d}$ are learnable weight matrices, $\odot$ is the Hadamard point-wise multiplication operator and the gate $\eta_{i,j}$ is defined as $\eta_{i,j} = \sigma\left(\mathbf{W}_3 \mathbf{h}_i^{(\ell-1)} + \mathbf{W}_4 \mathbf{h}_j^{(\ell-1)}\right)$, where $\mathbf{W}_3 \in \mathbb{R}^{2d \times 2d}$, $\mathbf{W}_4 \in \mathbb{R}^{2d \times 2d}$ are learnable weight matrices and with $\sigma$ as an activation function. In Bresson & Laurent (2017), the sigmoid function is chosen as the activation function so that the gate $\eta_{i,j}$ can learn the weight controlling how important the information from node $j$ to node $i$.

We concatenate the node representations of all layers to preserve the information of different-hop neighbors better. The concatenated representation of node $i$ after $n$ layers is $\mathbf{h}_i \in \mathbb{R}^{2(n+1)d}$,

$$\mathbf{h}_i = \text{CONCAT}\left(\left\{\mathbf{h}_i^{(\ell)} \middle| \ell = 0, 1, ..., n\right\}\right). \tag{4}$$

Note that our task is to estimate the similarity between two graphs. Therefore, we need to generate graph-level representations based on the node representations above. We design a *multi-scale pooling* technique that considers both attention pooling and summation pooling. Attention pooling (Bai et al., 2019) assigns weights to each node according to its importance, and then pools the node embeddings using a weighted sum based on the attention weights. Denote the resulting graph embedding as $\mathbf{z}_{\text{att}} \in \mathbb{R}^{2nd}$. And we get $\mathbf{z}_{\text{att}} = \sum_{i=1}^{|\mathcal{V}|} \sigma\left(\mathbf{h}_i^T \tanh\left(\frac{1}{|\mathcal{V}|} \mathbf{W}_5 \sum_{j=1}^{|\mathcal{V}|} \mathbf{h}_j\right)\right) \mathbf{h}_i$, where $\mathbf{W}_5 \in \mathbb{R}^{[2(n+1)d] \times [2(n+1)d]}$ is learnable and $\sigma$ is a sigmoid function. Summation pooling $\mathbf{z}_{\text{sum}} \in \mathbb{R}^{2(n+1)d}$ sums the node embeddings, i.e., $\mathbf{z}_{\text{sum}} = \sum_{i=1}^{|\mathcal{V}|} \mathbf{h}_i$.

We observe that both of the above pooling methods have some drawbacks: summation pooling treats all nodes equally, which may not be optimal; and attention in attention pooling runs the risk of overfitting on specific nodes. Therefore, in our multi-scale pooling, we mix these two pooling methods and let the model learn to trade off the two pooling operations, thus reducing the drawbacks of the two pooling methods. We denote the combined graph embedding $\mathbf{z} \in \mathbb{R}^{2(n+1)d}$ as follows:

$$\mathbf{z}_{\text{combined}} = \mathbf{a}\mathbf{z}_{\text{att}} + (1 - \mathbf{a})\mathbf{z}_{\text{sum}}, \tag{5}$$

where $\mathbf{a} \in \mathbb{R}^{2(n+1)d}$ is a vector that can be learned.

Similar to the node feature pre-processing, we pass the graph embedding that has gone through the pooling layer via an MLP to adjust its dimension to a suitable size for subsequent processing, and finally we get the graph embedding $\mathbf{z} \in \mathbb{R}^d$.

## 3.3 GED and MCS Prediction Objectives

After obtaining the graph embeddings $\mathbf{z}_1$ and $\mathbf{z}_2$ of two graphs $\mathcal{G}_1$ and $\mathcal{G}_2$, we now explain how to obtain predicted GED values and the design of training objective. The way to get MCS estimation and training objective naturally follows.

To get predicted GED, an intuitive idea is to compute the Euclidean distance between $\mathbf{z}_1$ and $\mathbf{z}_2$,

$$\text{distance}(\mathbf{z}_1, \mathbf{z}_2) = \|\mathbf{z}_1 - \mathbf{z}_2\|_2. \tag{6}$$

Moreover, inspired by Zhuo & Tan (2022), we introduce the Neural Tensor Network (NTN) (Socher et al., 2013) as a multi-headed weighted cosine similarity function to compute the interaction value

of the two graph embeddings in the capacity of a bias value as a complement to Eq. 6. NTN is a powerful method for quantifying relationships between representations. We denote the interaction value of embeddings $\mathbf{z}_1$ and $\mathbf{z}_2$ as:

$$\text{interaction}\left(\mathbf{z}_1, \mathbf{z}_2\right) = \text{MLP}\left(\text{ReLU}\left(\mathbf{z}_1^T \mathbf{W}_6^{[1:t]} \mathbf{z}_2 + \mathbf{W}_7 \text{CONCAT}\left(\mathbf{z}_1, \mathbf{z}_2\right) + \mathbf{b}\right)\right), \quad (7)$$

where $\mathbf{W}_6^{[1:t]} \in \mathbb{R}^{d \times d \times t}$ is a learnable weight tensor, $\mathbf{W}_7$ is a learnable weight matrix, $\mathbf{b} \in \mathbb{R}^t$ is a bias vector, $t$ is the hyperparameter controlling the NTN output and MLP($\cdot$) is a fully connected neural network that maps the similarity from $\mathbb{R}^t$ to $\mathbb{R}$.

Finally, our predicted GED value is

$$\text{GED}(\mathcal{G}_1, \mathcal{G}_2) = \boldsymbol{\beta}\text{distance}(\mathbf{z}_1, \mathbf{z}_2) + (1 - \boldsymbol{\beta})\text{interaction}(\mathbf{z}_1, \mathbf{z}_2), \quad (8)$$

where $\boldsymbol{\beta}$ is a scalar that can be learned.

Then we adopt the mean squared error between our predicted GED value $\text{GED}(\mathcal{G}_1, \mathcal{G}_2)$ and ground-truth GED value $\text{GED}^*(\mathcal{G}_1, \mathcal{G}_2)$, and have the loss function:

$$\mathcal{L} = \frac{1}{T} \sum_{(\mathcal{G}_1, \mathcal{G}_2) \in \mathcal{D} \times \mathcal{D}} \text{MSE}\left(\text{GED}(\mathcal{G}_1, \mathcal{G}_2), \text{GED}^*(\mathcal{G}_1, \mathcal{G}_2)\right), \quad (9)$$

where $T$ is the number of training graph pairs in a graph database $\mathcal{D}$ and $\text{GED}^*(\mathcal{G}_1, \mathcal{G}_2)$ is the ground-truth GED value between graph $\mathcal{G}_1$ and $\mathcal{G}_2$.

Eq. 8 can be generalized to other similarity metrics like MCS. Recall that MCS is a similarity metric measuring the largest common subgraph of two graphs. Therefore, we can consider the output of interaction, i.e., Eq. 7 as the similarity of the two graph embeddings and further consider Eq. 6 as a bias value. Therefore, we can keep the right-hand side of Eq. 8 unchanged and change the left-hand side to $\text{MCS}(\mathcal{G}_1, \mathcal{G}_2)$. The loss function in Eq. 9 follows for MCS.

## 4 ANALYSIS

We first prove that our method GRASP achieves high expressiveness and can pass 1-WL test, and then analyze the complexity of GRASP that is linear to the number of nodes in a graph pair.

We use the following proposition to formally state that under certain preconditions, our method can outperform the 1-WL test. We utilize the proof in Xu et al. (2019) that the the representation ability of 1-WL test is equivalent to that of standard MPGNNs in the graph isomorphism problem.

**Proposition 1.** *Given a pair of non-isomorphic graphs $\mathcal{G}_1$ and $\mathcal{G}_2$ that cannot be discriminated by the 1-WL test, and the two graphs have different sets of initial position encodings, then GRASP can generate different graph representations for the two non-isomorphic graphs.*

The preconditions of Proposition 1 include that two graphs should have different sets of initial position encodings. According to the definition of RWPE we use, nodes on two non-isomorphic graphs generally get different sets of RWPEs when $k$ is sufficiently large (Dwivedi et al., 2022). Thus, RWPE satisfies this precondition that the sets of positional encodings are different. Please see Appendix A.1 for detailed proof.

The inference time complexity of GRASP is linear to the number of nodes of the graph pair. Our node feature preprocessing module requires downscaling the dimensionality of the node features from $\mathbb{R}^m$ to $\mathbb{R}^d$, resulting in a time complexity of $O(md|\mathcal{V}|)$. The positional encoding module contains a random walk positional encoding pre-computation and an MLP. The pre-computation of random walk positional encoding takes $O(k|\mathcal{V}|^2)$ due to the sparse matrix multiplication. This pre-computation only needs to be calculated one time when the number of steps of the random walker $k$ is selected, and thus it is omitted for the complexity. The MLP($\cdot$) takes $O(kd|\mathcal{V}|)$ time. The node embedding module contains $n$ layers of RGGC with time complexity of $O(n|\mathcal{E}|)$. The multi-scale pooling module contains attention pooling and summation pooling, both with time complexity of $O(nd|\mathcal{V}|)$. In the phase of generating the final graph embedding, we downscale the dimensionality of the graph embedding from $\mathbb{R}^{2nd}$ to $\mathbb{R}^d$ with a time complexity of $O(nd^2)$. In the similarity prediction module, the time complexity of NTN interaction is $O(d^2t)$, where $t$ is the dimension of NTN output, and the time complexity of Euclidean distance calculation is $O(d)$ and the total time is $O(d^2t)$. Hence, the time complexity of GRASP for predictions is $O(d|\mathcal{V}|(m + k + n) + nd^2 + d^2t)$, which is linear to the number of nodes of the graph pair.

# 5 EXPERIMENTS

We evaluate our method GRASP against competitors on the GED and MCS prediction tasks.

## 5.1 EXPERIMENT SETUP

**Data and Ground Truth.** We conduct experiments on four real-world datasets, including AIDS700nef, LINUX, IMDBMulti (Bai et al., 2019) and PTC (Bai et al., 2020). Dataset descriptions and statistics can be found in Appendix A.2. We split training, validation, and testing data with ratio of 6:2:2 for all datasets and all methods by following the setting in Bai et al. (2019) and Bai et al. (2020). For the small datasets AIDS and LINUX, we use A* to calculate ground-truth GEDs. For IMDBMulti and PTC, we follow the way in Bai et al. (2019) and use the minimum of the results of three approximation methods Beam (Neuhaus et al., 2006), Hungarian (Riesen & Bunke, 2009) and VJ (Fankhauser et al., 2011) to be ground-truth GED. We use the MCSPLIT (McCreesh et al., 2017) algorithm to calculate the ground-truth MCS.

**Baseline Methods.** We compare with SIMGNN (Bai et al., 2019), GMN (Li et al., 2019), GRAPHSIM (Bai et al., 2020), MGMN (Ling et al., 2023), H2MN (Zhang et al., 2021), EGSC (Qin et al., 2021), ERIC (Zhuo & Tan, 2022), GREED (Ranjan et al., 2022). We use their official code provided by the authors and leave it unchanged for GED objective and simply extend it for MCS objective, since some methods are only implemented for GED. GENN-A* (Wang et al., 2021) cannot scale to large datasets, including IMDBMulti and PTC, and thus its result is omitted.

**Hyperparameters.** In our method, we use a search range of $\{$w/o, 8, 16, 24, 32$\}$ for the step size $k$ of the RWPE, $\{4, 6, 8, 10, 12\}$ for the number of layers $\ell$ of the GNN backbone, and $\{16, 32, 64, 128, 256\}$ for the dimensionality $d$ of the node hidden representations and also the final graph embedding. Our full hyperparameter settings on the four datasets and hyperparameter sensitivity analysis on the AIDS700nef dataset can be found in Appendix A.3 and A.4, respectively. For competitors, we conduct experiments according to their hyperparameter settings reported by their works.

**Evaluation metrics.** We compare the performance of all methods by Mean Squared Error (MSE), Spearman's Rank Correlation Coefficient ($\rho$) (Spearman, 1987), Kendall 's Rank Correlation Coefficient ($\tau$) (Kendall, 1938), and Precision at 10 and 20 (P@10 and 20). The lower MSE proves that the model performs better; the higher the last four, the better the model performs. All these metrics are widely used in previous studies (Bai et al., 2019; 2020). Note that, our method GRASP, same as the baseline GREED, is designed to predict GED directly, while the other methods predict normalized GED scores. Hence, on the acquisition of the MSE, to be consistent with other works including (Bai et al., 2019; 2020; Zhuo & Tan, 2022), we first normalize the outputs of GRASP and GREED by $s(\mathcal{G}_1, \mathcal{G}_2) = \exp\left(-\frac{2 \times \mathrm{GED}(\mathcal{G}_1, \mathcal{G}_2)}{|\mathcal{V}|_1 + |\mathcal{V}|_2}\right)$ for GED and by $s(\mathcal{G}_1, \mathcal{G}_2) = \frac{2 \times \mathrm{MCS}(\mathcal{G}_1, \mathcal{G}_2)}{|\mathcal{V}|_1 + |\mathcal{V}|_2}$ for MCS. We also evaluate the inference efficiency.

All experiments are carried out on a linux machine with Ubuntu system, CPU model Intel(R) Xeon(R) Gold 6226R CPU @ 2.90GHz and GPU model NVIDIA GeForce RTX 3090.

## 5.2 EFFECTIVENESS

Tables 1 and 2 report the overall effectiveness of all methods on all datasets for GED and MCS predictions, respectively. In Table 1, for GED predictions, our method GRASP outperforms all methods by all metrics on AIDS700nef and LINUX datasets, and by 4 out of 5 metrics on IMDB-Multi and PTC datasets. For example, On AIDS700nef, our method GRASP achieves high P@20 0.863, significantly outperforming the best competitor performance 0.78 of GREED by 10.6% relative improvement. On LINUX, our method GRASP significantly reduces MSE to 0.075, compared with the best competitor ERIC with 0.110 MSE.

In Table 2 for MCS predictions, our method GRASP outperforms all methods by all metrics on all four datasets. For example, On IMDBMulti, GRASP can achieve 0.965 for $\tau$ metric, which is much higher than the runner-up H2MN with 0.921. On PTC, the P@10 of GRASP is 0.681 while the best competitor EGSC has 0.563.

| | AIDS700nef | | | | | IMDBMulti | | | | |
|---|---|---|---|---|---|---|---|---|---|---|
| | MSE↓ | $\rho$↑ | $\tau$↑ | P@10↑ | P@20↑ | MSE↓ | $\rho$↑ | $\tau$↑ | P@10↑ | P@20↑ |
| SimGNN | $2.251_{\pm0.169}$ | $0.861_{\pm0.005}$ | $0.690_{\pm0.005}$ | $0.471_{\pm0.019}$ | $0.542_{\pm0.014}$ | $0.676_{\pm0.051}$ | $0.893_{\pm0.018}$ | $0.781_{\pm0.019}$ | $0.831_{\pm0.015}$ | $0.845_{\pm0.017}$ |
| GraphSim | $\underline{1.040}_{\pm0.031}$ | $0.841_{\pm0.003}$ | $0.683_{\pm0.004}$ | $0.417_{\pm0.017}$ | $0.499_{\pm0.011}$ | $1.275_{\pm0.117}$ | $0.877_{\pm0.009}$ | $0.781_{\pm0.012}$ | $0.728_{\pm0.021}$ | $0.760_{\pm0.012}$ |
| GMN | $2.692_{\pm0.079}$ | $0.762_{\pm0.003}$ | $0.662_{\pm0.004}$ | $0.399_{\pm0.017}$ | $0.476_{\pm0.013}$ | $4.702_{\pm0.672}$ | $0.691_{\pm0.010}$ | $0.608_{\pm0.109}$ | $0.589_{\pm0.109}$ | $0.551_{\pm0.081}$ |
| MGMN | $2.402_{\pm0.071}$ | $0.904_{\pm0.002}$ | $0.749_{\pm0.003}$ | $0.464_{\pm0.014}$ | $0.541_{\pm0.011}$ | $6.250_{\pm2.840}$ | $0.860_{\pm0.068}$ | $0.680_{\pm0.084}$ | $0.506_{\pm0.075}$ | $0.556_{\pm0.088}$ |
| H2MN | $1.044_{\pm0.062}$ | $0.871_{\pm0.002}$ | $0.719_{\pm0.003}$ | $0.475_{\pm0.010}$ | $0.561_{\pm0.010}$ | $\mathbf{0.410}_{\pm0.028}$ | $0.889_{\pm0.010}$ | $0.793_{\pm0.011}$ | $0.848_{\pm0.009}$ | $0.860_{\pm0.005}$ |
| EGSC | $1.637_{\pm0.099}$ | $0.896_{\pm0.003}$ | $0.733_{\pm0.005}$ | $0.592_{\pm0.012}$ | $0.650_{\pm0.014}$ | $0.689_{\pm0.195}$ | $\underline{0.934}_{\pm0.010}$ | $0.821_{\pm0.017}$ | $0.852_{\pm0.011}$ | $0.862_{\pm0.008}$ |
| ERIC | $1.467_{\pm0.030}$ | $0.903_{\pm0.001}$ | $0.761_{\pm0.005}$ | $0.608_{\pm0.035}$ | $0.654_{\pm0.006}$ | $\underline{0.451}_{\pm0.020}$ | $0.909_{\pm0.023}$ | $0.835_{\pm0.023}$ | $0.859_{\pm0.006}$ | $\underline{0.869}_{\pm0.003}$ |
| GREED | $1.432_{\pm0.059}$ | $\underline{0.913}_{\pm0.004}$ | $\underline{0.796}_{\pm0.004}$ | $\underline{0.710}_{\pm0.008}$ | $\underline{0.780}_{\pm0.007}$ | $1.174_{\pm0.094}$ | $0.930_{\pm0.007}$ | $\underline{0.865}_{\pm0.005}$ | $\underline{0.859}_{\pm0.004}$ | $0.858_{\pm0.002}$ |
| GraSP | $\mathbf{0.987}_{\pm0.017}$ | $\mathbf{0.930}_{\pm0.002}$ | $\mathbf{0.829}_{\pm0.002}$ | $\mathbf{0.806}_{\pm0.007}$ | $\mathbf{0.863}_{\pm0.010}$ | $0.789_{\pm0.110}$ | $\mathbf{0.940}_{\pm0.002}$ | $\mathbf{0.876}_{\pm0.005}$ | $\mathbf{0.868}_{\pm0.007}$ | $\mathbf{0.876}_{\pm0.003}$ |

| | LINUX | | | | | PTC | | | | |
|---|---|---|---|---|---|---|---|---|---|---|
| | MSE↓ | $\rho$↑ | $\tau$↑ | P@10↑ | P@20↑ | MSE↓ | $\rho$↑ | $\tau$↑ | P@10↑ | P@20↑ |
| SimGNN | $0.391_{\pm0.093}$ | $0.979_{\pm0.005}$ | $0.885_{\pm0.011}$ | $0.969_{\pm0.008}$ | $0.959_{\pm0.011}$ | $2.043_{\pm0.139}$ | $0.926_{\pm0.007}$ | $0.783_{\pm0.010}$ | $0.492_{\pm0.024}$ | $0.587_{\pm0.008}$ |
| GraphSim | $0.186_{\pm0.047}$ | $0.984_{\pm0.001}$ | $0.930_{\pm0.008}$ | $0.966_{\pm0.007}$ | $0.949_{\pm0.006}$ | $\mathbf{0.772}_{\pm0.033}$ | $0.941_{\pm0.004}$ | $0.828_{\pm0.007}$ | $0.510_{\pm0.068}$ | $0.600_{\pm0.065}$ |
| GMN | $1.596_{\pm0.217}$ | $0.924_{\pm0.007}$ | $0.788_{\pm0.006}$ | $0.783_{\pm0.009}$ | $0.773_{\pm0.010}$ | $2.210_{\pm0.432}$ | $0.661_{\pm0.004}$ | $0.652_{\pm0.012}$ | $0.244_{\pm0.042}$ | $0.392_{\pm0.020}$ |
| MGMN | $2.035_{\pm0.430}$ | $0.965_{\pm0.008}$ | $0.856_{\pm0.018}$ | $0.938_{\pm0.031}$ | $0.930_{\pm0.005}$ | $2.315_{\pm0.347}$ | $0.935_{\pm0.012}$ | $0.777_{\pm0.021}$ | $0.486_{\pm0.034}$ | $0.588_{\pm0.035}$ |
| H2MN | $0.882_{\pm0.147}$ | $0.977_{\pm0.002}$ | $0.899_{\pm0.004}$ | $0.948_{\pm0.006}$ | $0.922_{\pm0.008}$ | $1.913_{\pm0.269}$ | $0.913_{\pm0.012}$ | $0.767_{\pm0.013}$ | $0.500_{\pm0.014}$ | $0.595_{\pm0.003}$ |
| EGSC | $0.170_{\pm0.028}$ | $0.986_{\pm0.001}$ | $0.904_{\pm0.003}$ | $0.987_{\pm0.003}$ | $0.980_{\pm0.007}$ | $1.915_{\pm0.133}$ | $0.924_{\pm0.010}$ | $0.781_{\pm0.005}$ | $0.510_{\pm0.031}$ | $0.594_{\pm0.016}$ |
| ERIC | $\underline{0.110}_{\pm0.013}$ | $\underline{0.993}_{\pm0.002}$ | $\underline{0.968}_{\pm0.005}$ | $\underline{0.989}_{\pm0.004}$ | $\underline{0.981}_{\pm0.004}$ | $1.680_{\pm0.051}$ | $0.932_{\pm0.006}$ | $0.793_{\pm0.006}$ | $\underline{0.516}_{\pm0.012}$ | $\underline{0.605}_{\pm0.008}$ |
| GREED | $0.926_{\pm0.032}$ | $0.966_{\pm0.005}$ | $0.905_{\pm0.002}$ | $0.978_{\pm0.005}$ | $0.975_{\pm0.006}$ | $2.442_{\pm0.100}$ | $0.889_{\pm0.005}$ | $0.765_{\pm0.006}$ | $0.424_{\pm0.009}$ | $0.517_{\pm0.007}$ |
| GraSP | $\mathbf{0.075}_{\pm0.016}$ | $\mathbf{0.995}_{\pm0.002}$ | $\mathbf{0.971}_{\pm0.003}$ | $\mathbf{0.994}_{\pm0.002}$ | $\mathbf{0.991}_{\pm0.002}$ | $\underline{1.641}_{\pm0.131}$ | $\mathbf{0.946}_{\pm0.002}$ | $\mathbf{0.846}_{\pm0.002}$ | $\mathbf{0.602}_{\pm0.007}$ | $\mathbf{0.707}_{\pm0.011}$ |

Table 1: Effectiveness results on GED predictions with standard deviation. The MSE is in $10^{-3}$. **Bold**: best, Underline: runner-up.

| | AIDS700nef | | | | | IMDBMulti | | | | |
|---|---|---|---|---|---|---|---|---|---|---|
| | MSE↓ | $\rho$↑ | $\tau$↑ | P@10↑ | P@20↑ | MSE↓ | $\rho$↑ | $\tau$↑ | P@10↑ | P@20↑ |
| SimGNN | $6.148_{\pm0.261}$ | $0.838_{\pm0.004}$ | $0.666_{\pm0.004}$ | $0.406_{\pm0.017}$ | $0.484_{\pm0.011}$ | $0.494_{\pm0.157}$ | $0.976_{\pm0.002}$ | $0.912_{\pm0.004}$ | $0.866_{\pm0.013}$ | $0.893_{\pm0.012}$ |
| GraphSim | $3.601_{\pm0.325}$ | $0.805_{\pm0.020}$ | $0.647_{\pm0.023}$ | $0.358_{\pm0.038}$ | $0.428_{\pm0.029}$ | $3.530_{\pm0.241}$ | $0.897_{\pm0.013}$ | $0.839_{\pm0.018}$ | $0.674_{\pm0.023}$ | $0.706_{\pm0.023}$ |
| GMN | $3.817_{\pm0.488}$ | $0.829_{\pm0.009}$ | $0.613_{\pm0.007}$ | $0.433_{\pm0.032}$ | $0.492_{\pm0.016}$ | $2.351_{\pm0.312}$ | $0.891_{\pm0.011}$ | $0.779_{\pm0.024}$ | $0.646_{\pm0.094}$ | $0.682_{\pm0.032}$ |
| MGMN | $6.102_{\pm0.262}$ | $0.881_{\pm0.004}$ | $0.725_{\pm0.005}$ | $0.435_{\pm0.023}$ | $0.503_{\pm0.009}$ | $2.267_{\pm0.347}$ | $0.968_{\pm0.006}$ | $0.862_{\pm0.012}$ | $0.663_{\pm0.082}$ | $0.727_{\pm0.064}$ |
| H2MN | $3.301_{\pm0.253}$ | $0.822_{\pm0.008}$ | $0.665_{\pm0.009}$ | $0.370_{\pm0.017}$ | $0.442_{\pm0.015}$ | $\underline{0.214}_{\pm0.034}$ | $0.981_{\pm0.003}$ | $\underline{0.921}_{\pm0.007}$ | $0.847_{\pm0.010}$ | $0.874_{\pm0.009}$ |
| EGSC | $\underline{2.544}_{\pm0.213}$ | $\underline{0.915}_{\pm0.003}$ | $0.775_{\pm0.003}$ | $0.590_{\pm0.009}$ | $0.656_{\pm0.007}$ | $0.245_{\pm0.056}$ | $0.979_{\pm0.001}$ | $0.909_{\pm0.006}$ | $0.895_{\pm0.012}$ | $\underline{0.898}_{\pm0.018}$ |
| ERIC | $3.119_{\pm0.292}$ | $0.900_{\pm0.002}$ | $0.779_{\pm0.004}$ | $0.556_{\pm0.016}$ | $0.620_{\pm0.019}$ | $0.227_{\pm0.011}$ | $0.980_{\pm0.005}$ | $0.953_{\pm0.005}$ | $0.868_{\pm0.005}$ | $0.890_{\pm0.003}$ |
| GREED | $3.377_{\pm0.298}$ | $0.911_{\pm0.004}$ | $\underline{0.807}_{\pm0.005}$ | $\underline{0.774}_{\pm0.027}$ | $\underline{0.822}_{\pm0.013}$ | $1.472_{\pm0.460}$ | $0.869_{\pm0.006}$ | $0.782_{\pm0.009}$ | $\underline{0.887}_{\pm0.024}$ | $0.898_{\pm0.027}$ |
| GraSP | $\mathbf{2.120}_{\pm0.193}$ | $\mathbf{0.926}_{\pm0.002}$ | $\mathbf{0.830}_{\pm0.003}$ | $\mathbf{0.888}_{\pm0.017}$ | $\mathbf{0.908}_{\pm0.017}$ | $\mathbf{0.119}_{\pm0.032}$ | $\mathbf{0.985}_{\pm0.003}$ | $\mathbf{0.965}_{\pm0.004}$ | $\mathbf{0.927}_{\pm0.009}$ | $\mathbf{0.953}_{\pm0.006}$ |

| | LINUX | | | | | PTC | | | | |
|---|---|---|---|---|---|---|---|---|---|---|
| | MSE↓ | $\rho$↑ | $\tau$↑ | P@10↑ | P@20↑ | MSE↓ | $\rho$↑ | $\tau$↑ | P@10↑ | P@20↑ |
| SimGNN | $0.282_{\pm0.108}$ | $0.960_{\pm0.008}$ | $0.837_{\pm0.017}$ | $0.920_{\pm0.024}$ | $0.926_{\pm0.016}$ | $3.919_{\pm0.336}$ | $0.882_{\pm0.005}$ | $0.724_{\pm0.007}$ | $0.493_{\pm0.012}$ | $0.571_{\pm0.011}$ |
| GraphSim | $0.239_{\pm0.066}$ | $0.947_{\pm0.014}$ | $0.865_{\pm0.029}$ | $0.902_{\pm0.014}$ | $0.902_{\pm0.015}$ | $2.868_{\pm0.345}$ | $0.826_{\pm0.016}$ | $0.671_{\pm0.018}$ | $0.424_{\pm0.038}$ | $0.505_{\pm0.044}$ |
| GMN | $0.792_{\pm0.092}$ | $0.879_{\pm0.009}$ | $0.763_{\pm0.032}$ | $0.762_{\pm0.019}$ | $0.802_{\pm0.061}$ | $4.913_{\pm0.122}$ | $0.801_{\pm0.032}$ | $0.647_{\pm0.027}$ | $0.462_{\pm0.022}$ | $0.531_{\pm0.040}$ |
| MGMN | $1.244_{\pm0.201}$ | $0.902_{\pm0.015}$ | $0.772_{\pm0.026}$ | $0.775_{\pm0.042}$ | $0.814_{\pm0.048}$ | $4.293_{\pm0.271}$ | $\underline{0.922}_{\pm0.005}$ | $\underline{0.772}_{\pm0.008}$ | $0.480_{\pm0.017}$ | $0.560_{\pm0.014}$ |
| H2MN | $0.235_{\pm0.016}$ | $0.952_{\pm0.004}$ | $0.854_{\pm0.007}$ | $0.915_{\pm0.003}$ | $0.920_{\pm0.006}$ | $\underline{1.981}_{\pm0.048}$ | $0.871_{\pm0.006}$ | $0.711_{\pm0.008}$ | $0.486_{\pm0.014}$ | $0.568_{\pm0.016}$ |
| EGSC | $0.167_{\pm0.011}$ | $0.968_{\pm0.001}$ | $0.853_{\pm0.003}$ | $0.949_{\pm0.008}$ | $0.951_{\pm0.005}$ | $2.832_{\pm0.159}$ | $0.909_{\pm0.012}$ | $0.766_{\pm0.014}$ | $\underline{0.563}_{\pm0.045}$ | $0.633_{\pm0.027}$ |
| ERIC | $\underline{0.114}_{\pm0.010}$ | $\underline{0.971}_{\pm0.004}$ | $0.928_{\pm0.004}$ | $0.957_{\pm0.005}$ | $0.967_{\pm0.007}$ | $3.188_{\pm0.240}$ | $0.880_{\pm0.008}$ | $0.732_{\pm0.011}$ | $0.518_{\pm0.016}$ | $0.582_{\pm0.015}$ |
| GREED | $0.280_{\pm0.110}$ | $0.812_{\pm0.002}$ | $0.719_{\pm0.002}$ | $\underline{0.974}_{\pm0.009}$ | $\underline{0.982}_{\pm0.005}$ | $5.915_{\pm1.215}$ | $0.841_{\pm0.016}$ | $0.714_{\pm0.019}$ | $0.541_{\pm0.052}$ | $\underline{0.637}_{\pm0.041}$ |
| GraSP | $\mathbf{0.064}_{\pm0.020}$ | $\mathbf{0.977}_{\pm0.002}$ | $\mathbf{0.939}_{\pm0.002}$ | $\mathbf{0.978}_{\pm0.005}$ | $\mathbf{0.985}_{\pm0.004}$ | $\mathbf{1.922}_{\pm0.052}$ | $\mathbf{0.927}_{\pm0.004}$ | $\mathbf{0.795}_{\pm0.007}$ | $\mathbf{0.681}_{\pm0.010}$ | $\mathbf{0.718}_{\pm0.021}$ |

Table 2: Effectiveness results on MCS predictions with standard deviation. The MSE is in $10^{-3}$. **Bold**: best, Underline: runner-up.

The overall results in Tables 1 and 2 demonstrate the superior power of GraSP with simple but effective designs, including positional encoding enhanced node features and multi-scale pooling on RGGC backbone.

To visualize the superiority of our method, we compare our approach GraSP on the absolute GED error with the recent baselines GREED and ERIC by absolute GED error heatmaps as shown in Appendix A.5, which intuitively shows that GraSP can reduce prediction errors over existing methods.

## 5.3 EFFICIENCY

We report the inference time per 10k graph pairs of all approaches on every dataset in Figure 4. Our method GraSP is the fastest method on all datasets to complete the inference. The efficiency of GraSP is due to the following designs. First, GraSP does not need the expensive cross-graph node-level interactions that are usually adopted in existing methods. Second, the positional encoding used in GraSP to enhance node features can be precomputed and reused for the efficiency of online

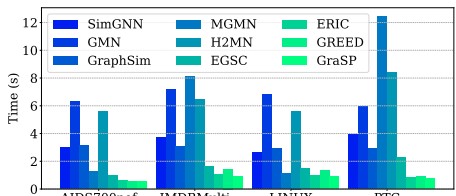

Figure 4: Inference time in second(s) per 10k pairs.

| | MSE | $\rho$ | $\tau$ | P@10 | P@20 |
|---|---|---|---|---|---|
| GRASP (GIN) | 1.092 | 0.921 | 0.815 | 0.761 | 0.831 |
| GRASP (GCN) | 1.304 | 0.909 | 0.796 | 0.717 | 0.786 |
| GRASP (w/o pe) | 1.101 | 0.920 | 0.812 | 0.772 | 0.838 |
| GRASP (w/o att) | 1.089 | 0.923 | 0.817 | 0.789 | 0.844 |
| GRASP (w/o sum) | 1.077 | 0.922 | 0.817 | 0.779 | 0.847 |
| GRASP (w/o NTN) | 1.473 | 0.915 | 0.802 | 0.728 | 0.805 |
| GRASP | **0.987** | **0.930** | **0.829** | **0.806** | **0.863** |

Table 3: Ablation study on AIDS700nef under GED metric. MSE is in $10^{-3}$.

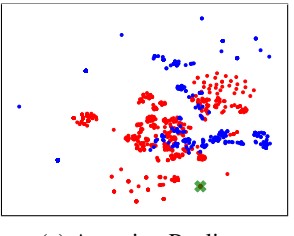
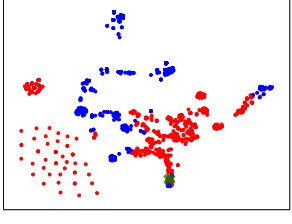
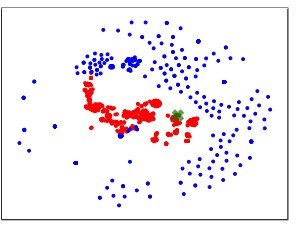

| (a) Attention Pooling. | (b) Add Pooling. | (c) Multi-scale Pooling. |
|---|---|---|

Figure 5: T-SNE visualization on IMDBMulti. We plot the graph embeddings on a 2d plain, where the green cross denotes a randomly chosen query graph, red dots denote the top fifty percent of similar graphs in the database and blue dots denote the later fifty percent.

inference. Third, all the technical components in GRASP are designed to make the complicated simple and effective for graph similarity predictions, resulting in the efficient performance. The inference time of SIMGNN, GMN, GRAPHSIM and MGMN is longer, which is due to the expensive cross-graph node-level interactions that these models explicitly perform during inference. EGSC, ERIC, and GREED are relatively faster but do not exceed our method.

## 5.4 ABLATION STUDY

In the ablation study, we first compare GRASP with RGGC backbone over GRASP with GCN and GIN backbones. We also ablate the positional encoding in Section 3.1, the attention pooling and summation polling in the multi-scale pooling in Section 3.2 of GRASP, and the NTN in Section 3.3 denoted as w/o pe, w/o att, w/o sum, and w/o NTN respectively. The results on AIDS700nef for GED are reported in Table 3. Observe that GRASP obtains the best performance on all metrics than all its ablated versions, which proves the effectiveness of all our proposed components in GRASP. In particular, with only either attention or summation pooling, the performance is inferior to GRASP with the proposed multi-scale pooling technique that hybrids both pooling techniques, which validates the rationale for designing the technique.

To further exemplify the effect of multi-scale pooling, we conducted experiments on IMDBMulti dataset with t-SNE visualization (van der Maaten & Hinton, 2008), as shown in Figure 5. Compared to the graph embeddings obtained using only attention and summation pooling methods, the graph embeddings obtained using multi-scale pooling technique exhibit better patterns in the embedding space, which reflects the effective modeling of graph similarity properties.

## 5.5 CASE STUDY

We conduct a case study of GRASP over drug discovery. Graph similarity search is often applied to drug discovery with an essential role in the precursor step before final molecular screening (Ranu et al., 2011).The molecules identified by graph similarity are crucial to the quality of the final screen. In this case study, to examine the application of our method, we consider the ranking of returned graphs with respect to a query graph. In Figure 6, we show a case of ranking under the GED metric on AIDS700nef dataset. Specifically, given a query graph on the left side of Figure 6, the first row shows the ground-truth ranking of graphs with GED to the query, and the second row shows the graphs ranked by predicted GED from our method GRASP. As shown, GRASP is able to accurately predict the GED values and rank the graphs with similar structures to the top, and the ranking from 1 to 560 are almost the same as the ground truth. We further provide the case studies on the other three datasets in Appendix A.6.

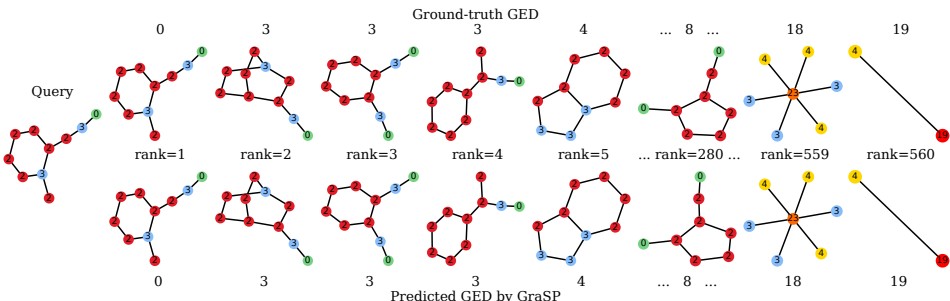

Figure 6: A ranking case study of GED prediction on AIDS700nef.

# 6 RELATED WORK

Exact calculation of GED and MCS is an NP-hard problem. For GED, recent works (Kim et al., 2019; Kim, 2020; Chang et al., 2020) compress the search space for faster filtering and verification, but still remain inefficient. For MCS, recent work (McCreesh et al., 2017) introduces a branch and bound algorithm that compresses the memory and computational requirements of the search process. Approximation methods like Beam (Neuhaus et al., 2006), Hungarian (Riesen & Bunke, 2009) and VJ (Fankhauser et al., 2011) employ heuristic search and trade precision for reduced complexity, but still have sub-exponential or cubical cost. Wang et al. (2021) uses combinatorial technique to combine heuristic and learning methods, but still does not scale well (Ranjan et al., 2022).

In recent years, many learning-based methods have emerged which achieve accurate prediction of similarity values between graphs, while allowing the use of fewer computational resources. SIMGNN (Bai et al., 2019) adopts a Siamese network structure, an NTN module to compare graph embeddings, and then also uses histogram features of node embeddings to capture fine-grained node-level comparison information. GMN (Li et al., 2019) contains an attention module for cross-graph node matching and then encodes cross-graph node matching information into node embeddings to solve for graph similarity. GRAPHSIM (Bai et al., 2020) uses a CNN to capture multi-scale node-level interactions. MGMN (Ling et al., 2023) introduces a node-graph matching layer to capture interactions across levels (between nodes and graphs). H2MN Zhang et al. (2021) uses the concept of hypergraphs; after constructing hypergraphs, each hyperedge after pooling in a hypergraph is used as a subgraph for matching between subgraphs. These methods explicitly use the cross-graph node-level interactions. ERIC (Zhuo & Tan, 2022) proposes a soft matching module aiming to be used during training while to be removed when inference to speed up inference time. However, the cross-graph node-level interaction module may not be necessary, while simple but effective designs can already achieve superior performance. EGSC (Qin et al., 2021) uses the knowledge distillation method to extract the knowledge learned by the teacher model to get a lighter-weight student model, but the upper limit of the performance of the student model does not exceed that of the teacher model. The latest GREED (Ranjan et al., 2022) proposes the concept of pair-independent embeddings that can be indexed, and further addresses the issue of cross-graph interactions. Our model includes a novel embedding structure that incorporates a positional encoding technique, as well as a new multi-scale pooling technique, to be more expressive than the 1-WL test to improve model performance in graph similarity/distance prediction tasks.

# 7 CONCLUSION

In this paper, we present GRASP, a simple but effective and efficient method for accurate predictions on GED and MCS, two important graph similarity metrics with a wide range of applications in various fields. To make the complicated simple, we design a series of rational and effective techniques in GRASP to achieve superior performance. In particular, we design techniques to enhance node features via positional encoding, employ a robust graph neural network, and develop a multi-scale pooling technique. We theoretically prove that our method is more expressive and passes 1-WL test. In extensive experiments, GRASP is versatile in predicting GED and MCS metrics accurately on real-world datasets, often outperforming existing methods by a significant margin.

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

# A APPENDIX

## A.1 PROOFS OF PROPOSITIONS

**Proof of Proposition 1.**

*Proof.* Suppose that the 1-WL test still fails to distinguish between $\mathcal{G}_1$ and $\mathcal{G}_2$ through $n$ iterations. This is equivalent to the fact that the set of node representations $\{\mathbf{h}_u^{\ell}\}$ generated is the same for any layer $\ell$ from the 1-st to the n-th layer of the standard MPGNN; hence, the final graph representations are also the same.

At $\ell = 0$, it is clear that the set of node label features of two graphs $\{\boldsymbol{\mu}_u\}$ is the same, and since the two feature sets $\{\mathbf{p}_u | u \in \mathcal{G}_1\}$ and $\{\mathbf{p}_v | v \in \mathcal{G}_2\}$ are not the same, according to Eq. 2, where positional coding has been used to enhance the initial node representations, the sets of node representations at $\ell = 0$ of the two graphs, $\{\mathbf{h}_u^{(0)}\} = \{\text{CONCAT}(\boldsymbol{\mu}_u, \mathbf{p}_u)\}$ and $\{\mathbf{h}_v^{(0)}\}$ are not the same. Therefore, due to the usage of node concatenated representations in Eq. 4, the sets of concatenated representations $\{\mathbf{h_u}\}$ and $\{\mathbf{h_v}\}$ is not the same. Therefore, the final two-graph embeddings are different. □

## A.2 DESCRIPTIONS AND STATISTICS OF THE DATASETS

**AIDS700nef.** The AIDS[1] dataset consists of compounds that exhibit anti-HIV properties after screening. A total of 700 compounds of which less than or equal to 10 nodes were selected by (Bai et al., 2019) to form the AIDS700nef dataset. There are 29 node labels in the AIDS700nef. **IMDBMulti.** The IMDBMulti (Yanardag & Vishwanathan, 2015) is a movie collaboration dataset, where nodes represent an actor and edges indicate whether two actors appear in the same movie. **LINUX.** The LINUX dataset consists of a series of Program Dependency Graphs (PDGs) generated by (Wang et al., 2012), where a node denotes a statement and an edge denotes a dependency between two statements. The LINUX dataset we used in our experiments consists of 1000 graphs randomly selected by (Bai et al., 2019) in the original LINUX dataset. **PTC.** The PTC (Toivonen et al., 2003) dataset contains a series of compounds labeled according to their carcinogenicity in male and female mice and rats. There are 19 node labeles in the PTC. The detailed statics of the four dataset are listed in Table 4.

| Datasets | # Graphs | # Pairs | # Features | Avg # Nodes |
|---|---|---|---|---|
| AIDS700nef | 700 | 78400 | 29 | 8.9 |
| IMDBMulti | 1500 | 360000 | 1 | 13.0 |
| LINUX | 1000 | 160000 | 1 | 7.6 |
| PTC | 344 | 18975 | 19 | 25.6 |

Table 4: Statistics of datasets.

## A.3 HYPERPARAMETER SETTINGS

We list our 8 hyperparameter settings in Table 5.

## A.4 SENSITIVITY ANALYSIS

We evaluate how the step size $k$ of the RWPE, the number of layers $\ell$ of the GNN backbone and the dimensionality $d$ of the node's hidden representations will affect on the performance on the AIDS700nef dataset. We list the MSE values of the AIDS700nef on Figure 7. We find that the model achieves optimal performance with our hyperparameter settings $k = 16$, $\ell = 8$ and $d = 64$. We can also observe that the use of RWPE is improving the performance, and the performance first improves and then stabilizes after $k > 16$. This is due to the fact that the average number of nodes in AIDS700nef is less than 10, and also that the properties of RWPE that we utilize ensures that when $k$ is large enough, it can be guaranteed that the two non-isomorphic graphs have different sets of RWPEs. When $l$ and $d$ are too large, the performance will drop because of overfitting.

---

[1]https://wiki.nci.nih.gov/display/NCIDTPdata/AIDS+Antiviral+Screen+Data.

| | Params | AIDS700nef | IMDBMulti | LINUX | PTC |
|---|---|---|---|---|---|
| GED | learning rate | $1e^{-4}$ to $1e^{-3}$ | $1e^{-4}$ to $1e^{-3}$ | $2e^{-4}$ to $2e^{-3}$ | $1e^{-4}$ to $1e^{-3}$ |
| | weight decay | $5e^{-4}$ | $5e^{-4}$ | $5e^{-4}$ | $5e^{-4}$ |
| | batch size | 256 | 256 | 256 | 256 |
| | epochs | $3e^4$ | $3e^4$ | $2e^4$ | $5e^3$ |
| | # of gnn layers | 8 | 4 | 8 | 8 |
| | hidden dims. | 64 | 64 | 64 | 64 |
| | RWPE dims. | 16 | 16 | 10 | 20 |
| MCS | learning rate | $1e^{-4}$ to $1e^{-3}$ | $1e^{-4}$ to $1e^{-3}$ | $2e^{-4}$ to $2e^{-3}$ | $1e^{-4}$ to $1e^{-3}$ |
| | weight decay | $5e^{-4}$ | $5e^{-4}$ | $5e^{-4}$ | $5e^{-4}$ |
| | batch size | 256 | 256 | 256 | 256 |
| | epochs | $3e^4$ | $3e^4$ | $2e^4$ | $2e^4$ |
| | # of gnn layers | 8 | 3 | 8 | 8 |
| | hidden dims. | 64 | 64 | 64 | 64 |
| | RWPE dims. | 16 | 16 | 10 | 20 |

Table 5: Hyperparamater settings on 4 datasets.

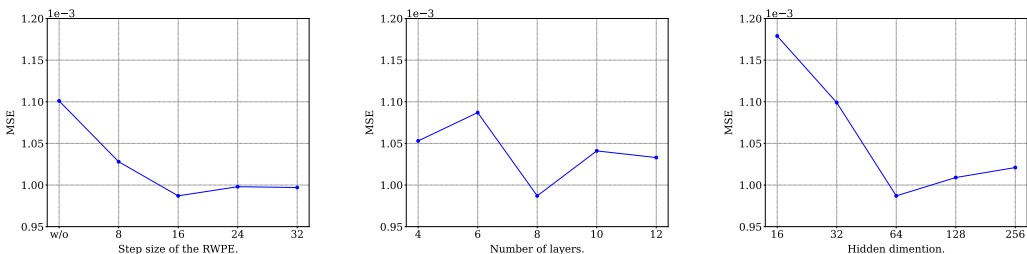

Figure 7: Hyperparameters sensitivity analysis.

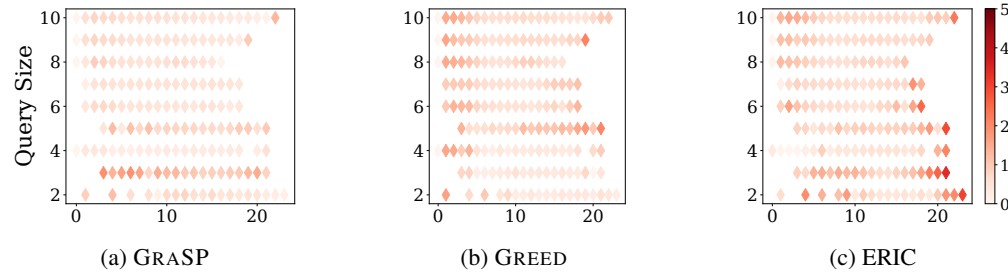

(a) GRASP      (b) GREED      (c) ERIC

Figure 8: Absolute error heatmap on GED on AIDS700nef.

### A.5 A COMPARISON ON GED PREDICTION ERROR HEATMAP

The absolute GED error heatmaps of our approach, GREED and ERIC on four datasets are shown in Figure 8, 9, 10 and 11. The x-axis represents the GED between the query graph and the target graphs. The y-axis represents the number of nodes in the query graph. The color of each dot represents the absolute error on GED between a query graph and a target graph. The lighter color indicates the lower absolute error. Our method GRASP has better performance than existing methods over different query sizes and GED values on all datasets.

### A.6 ADDITIONAL CASE STUDIES

Three case studies on IMDBMulti, LINUX and PTC under GED metric are included in Figure 12, 13 and 14. Note that for AIDS700nef and LINUX dataset, the exact GEDs are obtained by the A* algorithm and the predicted results are obtained by our model. For the IMDBMulti and PTC dataset,

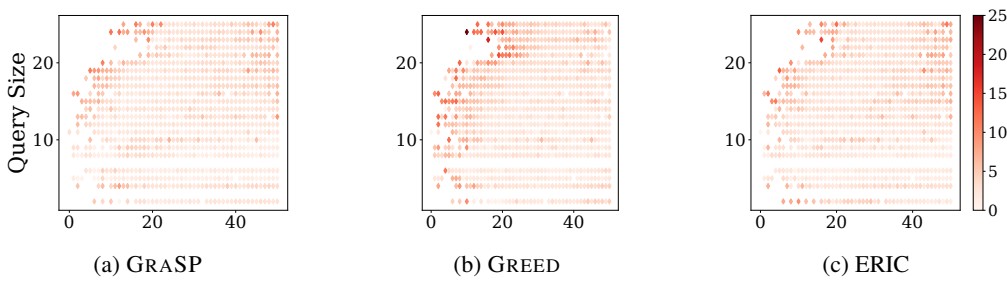

Figure 9: Absolute error heatmap on GED on PTC.

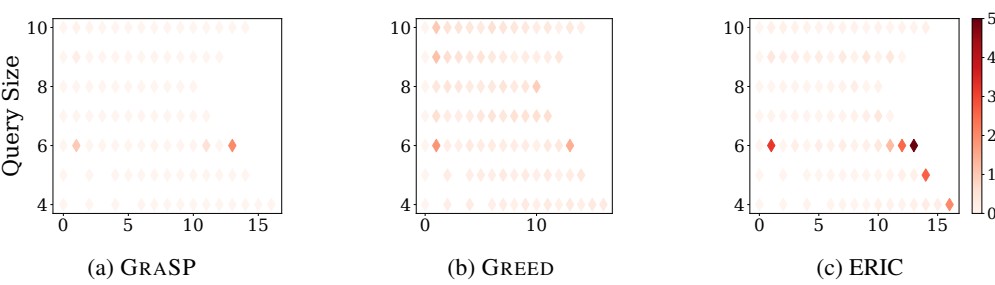

Figure 10: Absolute error heatmap on GED on LINUX.

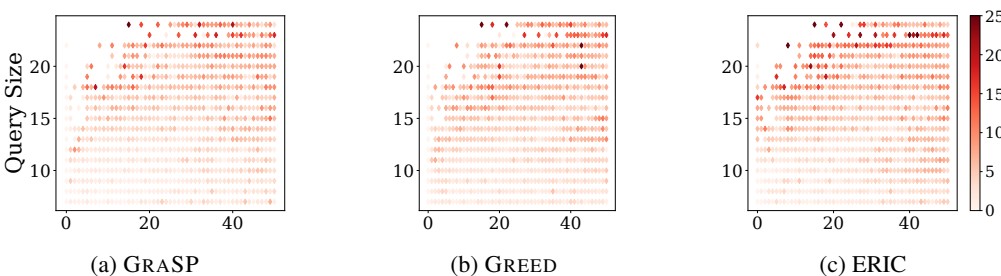

Figure 11: Absolute error heatmap on GED on IMDBMulti.

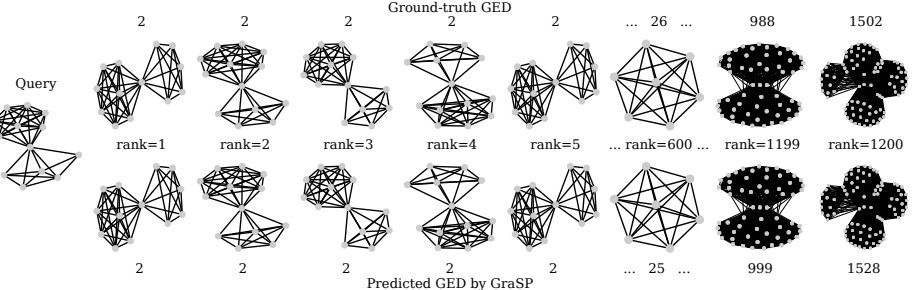

Figure 12: A ranking case on IMDBMulti.

we use the minimum of the calculated results for Beam, Hungarian, and VJ as the ground-truth GEDs due to infeasibility of computing the exact GEDs of these relatively large graphs.

## A.7 RESULTS ON PREDICTING GED DIRECTLY

Except GRASP and GREED, all the remaining baselines predict the exponentially normalized GEDs, i.e, similarity scores. In order to directly predict GED, we modify the official codes of all baselines

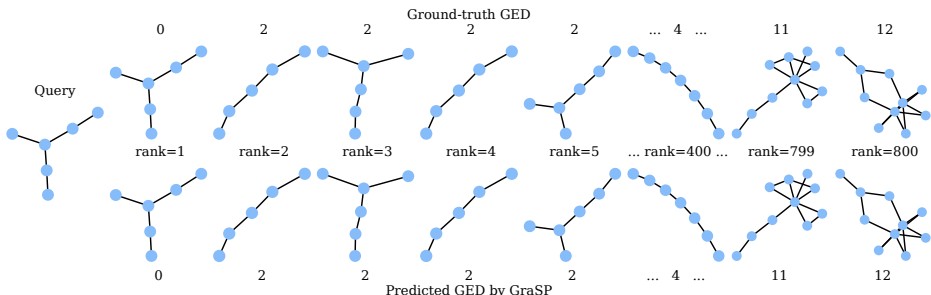

Figure 13: A ranking case on LINUX.

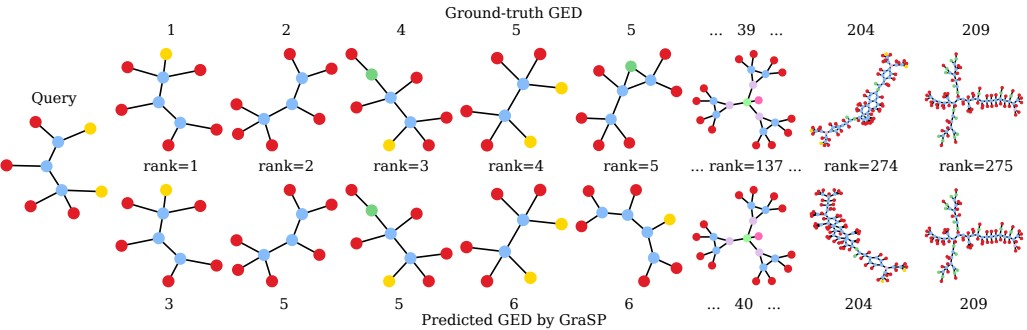

Figure 14: A ranking case on PTC.

| | AIDS700nef | | | | | IMDBMulti | | | | |
|---|---|---|---|---|---|---|---|---|---|---|
| | RMSE ↓ | $\rho$ ↑ | $\tau$ ↑ | P@10 ↑ | P@20 ↑ | RMSE ↓ | $\rho$ ↑ | $\tau$ ↑ | P@10 ↑ | P@20 ↑ |
| SIMGNN | 1.008 | 0.856 | 0.680 | 0.519 | 0.631 | 18.041 | 0.925 | 0.811 | 0.696 | 0.696 |
| GRAPHSIM | 1.153 | 0.803 | 0.657 | 0.364 | 0.499 | 31.372 | 0.728 | 0.619 | 0.575 | 0.571 |
| GMN | 0.947 | 0.870 | 0.695 | 0.579 | 0.579 | 13.027 | 0.922 | 0.803 | 0.823 | 0.827 |
| MGMN | 1.015 | 0.896 | 0.762 | 0.539 | 0.650 | 29.887 | 0.925 | 0.790 | 0.683 | 0.733 |
| H2MN | 0.924 | 0.877 | 0.745 | 0.557 | 0.653 | 25.401 | 0.900 | 0.798 | 0.781 | 0.790 |
| EGSC | 0.946 | 0.902 | 0.740 | 0.693 | 0.757 | 9.194 | 0.941 | 0.852 | 0.850 | 0.877 |
| ERIC | 0.971 | 0.880 | 0.751 | 0.599 | 0.688 | 9.032 | 0.934 | 0.873 | 0.812 | 0.832 |
| GREED | 0.884 | 0.899 | 0.776 | 0.661 | 0.741 | 8.171 | 0.933 | 0.867 | 0.857 | 0.866 |
| GRASP | **0.808** | **0.923** | **0.810** | **0.754** | **0.819** | **7.924** | **0.946** | **0.891** | **0.867** | **0.881** |

| | LINUX | | | | | PTC | | | | |
|---|---|---|---|---|---|---|---|---|---|---|
| | RMSE ↓ | $\rho$ ↑ | $\tau$ ↑ | P@10 ↑ | P@20 ↑ | RMSE ↓ | $\rho$ ↑ | $\tau$ ↑ | P@10 ↑ | P@20 ↑ |
| SIMGNN | 0.518 | 0.931 | 0.779 | 0.852 | 0.864 | 8.364 | 0.903 | 0.772 | 0.420 | 0.577 |
| GRAPHSIM | 0.232 | 0.970 | 0.897 | 0.974 | 0.982 | 11.659 | 0.848 | 0.710 | 0.401 | 0.519 |
| GMN | 0.369 | 0.946 | 0.804 | 0.968 | 0.966 | 7.551 | 0.901 | 0.774 | 0.525 | 0.633 |
| MGMN | 0.587 | 0.970 | 0.889 | 0.968 | 0.949 | 6.767 | 0.942 | 0.839 | 0.393 | 0.512 |
| H2MN | 0.559 | 0.963 | 0.876 | 0.945 | 0.945 | 8.090 | 0.880 | 0.758 | 0.408 | 0.539 |
| EGSC | 0.215 | 0.949 | 0.813 | 0.985 | 0.990 | 6.882 | 0.895 | 0.769 | 0.530 | 0.624 |
| ERIC | 0.285 | 0.971 | 0.917 | 0.977 | 0.989 | 6.401 | 0.881 | 0.735 | 0.409 | 0.529 |
| GREED | 0.414 | 0.966 | 0.902 | 0.969 | 0.978 | 4.970 | 0.921 | 0.810 | 0.455 | 0.569 |
| GRASP | **0.137** | **0.975** | **0.923** | **0.986** | **0.992** | **4.826** | **0.948** | **0.852** | **0.597** | **0.701** |

Table 6: Effectiveness results on original GED predictions. **Bold**: best, Underline: runner-up.

and train them by original GED. Table 6 reports the overall effectiveness of all methods on all datasets for GED predictions. RMSE represents root mean square error. Observe that our method GRASP consistently outperforms existing methods, while GREED and ERIC are with top performance.

## A.8 EXTENDING TO CONSIDER OF EDGE RELABELING

Here we discuss how to extend GRASP to include the cost of relabeling edges. Following (Dwivedi et al., 2023), Eq. 3 can be modified as:

$$\mathbf{h}_i^{(\ell)} = \mathbf{h}_i^{(\ell-1)} + \text{ReLU}\left(\mathbf{W}_1\mathbf{h}_i^{(\ell-1)} + \sum_{j \in \mathcal{N}(i)} \mathbf{e}_{i,j}^{(\ell)} \odot \mathbf{W}_2\mathbf{h}_j^{(\ell-1)}\right), \tag{10}$$

where $\mathbf{e}_{i,j}^{(\ell)}$ is the edge gate and can be defined as:

$$\mathbf{e}_{i,j}^{(\ell)} = \sigma\left(\mathbf{e}_{i,j}^{(\ell-1)} + \text{ReLU}\left(\mathbf{A}\mathbf{h}_i^{(\ell-1)} + \mathbf{B}\mathbf{h}_j^{(\ell-1)} + \mathbf{C}\mathbf{e}_{i,j}^{(\ell-1)}\right)\right), \tag{11}$$

where $\mathbf{A}, \mathbf{B}, \mathbf{C} \in \mathbb{R}^{2d \times 2d}$ and $\mathbf{e}_{i,j}^{(0)}$ represents the input edge feature.

## A.9 GENERALIZATION ABILITY TO LARGE UNSEEN GRAPHS

To test the generalization ability of GRASP on large unseen graphs for GED predictions, we conduct experiments by following the setting in (Ranjan et al., 2022). Specifically, we get GRASP-25 and GRASP-50 by training GRASP on graphs with node sizes up to 25 and 50, respectively, and then test on graph pairs in which the number of nodes in the query graph is in the range [25, 50]. Table 7 shows the results, with comparison to GREED and H2MN. Observe that (i) the performance of all methods degrades for large query sizes in [25,50], compared with the entire set in [0,50], (ii) when trained using smaller graphs from 50 to 25 size, all methods also degrade, (iii) our method GRASP-25 (resp. GRASP-50) keeps the best performance than the baselines under all these generalization settings.

|  | Query Size in [0,50] | Query Size in [25, 50] |
|---|---|---|
| GRASP-50 | **5.418** | **6.104** |
| GREED-50 | 5.885 | 7.470 |
| H2MN-50 | 7.041 | 9.024 |
| GRASP-25 | **7.499** | **9.841** |
| GREED-25 | 8.341 | 10.204 |
| H2MN-25 | 8.965 | 10.956 |

Table 7: Generalization on large unseen graphs with RMSE results to predict GED directly on PTC.

