# OpenReview forum: "GraSP: Simple yet Effective Graph Similarity Predictions"
_ICLR.cc/2024/Conference — Submitted to ICLR 2024_

### Official Review · Reviewer_dX6K · 2023-10-22

**Soundness:** 4 excellent
**Presentation:** 1 poor
**Contribution:** 2 fair
**Rating:** 5
**Confidence:** 3

**Summary:**

The paper introduces a novel approach, GRASP, for predicting graph similarity, specifically focusing on GED and MCS metrics. GRASP deviates from the trend of incorporating complex mechanisms by introducing a simplified model that utilizes positional encoding and RGGC to enhance the expressiveness and efficiency of graph neural networks. The authors claim theoretical superiority over the 1-WL test, a widely recognized method for graph isomorphism.

**Strengths:**

S1: The authors innovatively incorporate positional encoding within GNN framework, a commendable step that advances the GNN's ability to capture nuanced structural information. This ingenuity potentially sets a new precedent for subsequent research in graph similarity assessment.\
S2: The methodology introduced in this paper demonstrates notable efficiency. \
S3: The experimental results seem to be promising.

**Weaknesses:**

W1: The presentation of the content, particularly in Section 3, lacks clarity and cohesiveness, making it challenging for readers to follow and understand the proposed methodology. \
W2: The paper posits inefficiency in contemporary cross-graph interaction techniques as a primary catalyst for the development of GRASP. However, the narrative lacks a coherent demonstration of how GRASP mitigates these inefficiencies. The empirical section, intended to validate the method's enhanced efficiency, does not decisively support this assertion. Specifically, the performance metrics juxtaposed with existing strategies such as GREED and ERIC suggest comparable efficiencies, an outcome that muddles the purported superiority of GRASP in this domain. The authors should consider a more nuanced exposition of the method's unique efficiencies, supplemented by robust experimental evidence, to substantiate claims of its advancement over current practices. \
W3: The authors assert that prevailing cross-graph interaction modules contribute significantly to computational overheads. However, the delineation of how their proposed GRASP framework, which ostensibly employs similar cross-graph interactions via NTN, innovates upon or diverges from traditional methodologies is ambiguous. This lack of clarity muddles the reader's understanding of any novel contributions the paper might be making in this specific aspect of the framework. It is imperative for the authors to elucidate the nuanced operational differences, if any, introduced by GRASP that ameliorate the time-costly nature of cross-graph interactions, distinctly setting their approach apart from conventional ones. This clarification could significantly enhance the perceived value and ingenuity of their methodology.

**Questions:**

Q1: Numerous methods exist to enhance the expressiveness of GNNs. What motivated your decision to exclusively focus on positional encoding in your approach? \
Q2: What is the rationale behind using RGGC as a backbone? Furthermore, can you explain how the gating mechanism contributes to the effectiveness of your task? \
Q3: On page 4 of your paper, you note that "both of the above pooling methods have some drawbacks." Can you offer more specific evidence or instances that highlight these limitations? \
Q4: Could you delve into the key differences between GED and MCS, explaining the importance of considering MCS when many related studies concentrate primarily on GED?
Q5: What sets NTN apart from existing methods of cross-graph interaction, and why is NTN a suitable option for your proposed approach? \
Q6: How does GRASP tackle the issue of efficiency, and what factors make it an efficient solution?

---

> ### Author Response · Authors · 2023-11-20
> **Response to Reviewer dX6K and Look Forward to Your Reply (1/2)**
>
> We thank your insightful comments, and we have thoroughly addressed them as follows. We look forward to your reply.
>
> > **W1.** The presentation of the content, particularly in Section 3, lacks clarity and cohesiveness.
>
> **Response:** Thanks for the suggestion. We have enhanced the overview paragraph at the beginning of Section 3 to make it clear and cohesive.
>
>
> > **W2.** The authors should consider a more nuanced exposition of the method's unique efficiencies, supplemented by robust experimental evidence, to substantiate claims of its advancement over current practices.
> **Q6.** How does GRASP tackle the issue of efficiency, and what factors make it an efficient solution?
>
> **Response:** The efficiency of the proposed GraSP is due to the following designs. First, GraSP does not need the expensive cross-graph node-level interactions that are usually adopted in existing methods. Second, the positional encoding used in GraSP to enhance node features can be precomputed and reused for the efficiency of online inference. Third, all the technical components in GraSP are designed to make the complicated simple and effective for graph similarity predictions, resulting to efficient performance. By following the literature, we report the inference time per 10K pairs in *Figure 4* of the paper, where GraSP is the fastest method to predict graph similarity scores, which experimentally validates the efficiency of GraSP.
> As suggested, we have added the exposition above into Section 5.3 of the revised paper.
>
>
> > **W3.** The authors assert that prevailing cross-graph interaction modules contribute significantly to computational overheads. However, the delineation of how their proposed GRASP framework, which ostensibly employs similar cross-graph interactions via NTN, innovates upon or diverges from traditional methodologies is ambiguous. This lack of clarity muddles the reader's understanding of any novel contributions the paper might be making in this specific aspect of the framework. It is imperative for the authors to elucidate the nuanced operational differences, if any, introduced by GRASP that ameliorate the time-costly nature of cross-graph interactions, distinctly setting their approach apart from conventional ones. This clarification could significantly enhance the perceived value and ingenuity of their methodology.
>
> **Response:** We clarify that in Eq. (7), NTN is applied to two graph embeddings $\mathbf{z}_1$ and $\mathbf{z}_2$ to get an interaction value, interaction$(\mathbf{z}_1, \mathbf{z}_2)$. The interaction value in Eq. (7) is essentially a multi-headed weighted cosine similarity score, which is efficient to compute. It is *different* from the cross-graph *node-level* interactions required in existing studies as reviewed in Section 6. Specifically, the interactions in existing studies refer to the expensive node-level matching or node-graph matching processes across graphs, which take quadratic time w.r.t. the number of nodes.
> As explained in the response to **W2** above, our GraSP does not need such expensive cross-graph nodel-level interactions.
>
> > **Q5.** What sets NTN apart from existing methods of cross-graph interaction, and why is NTN a suitable option for your proposed approach?
>
> **Response:** Following the response to W3 above, NTN is a powerful way to quantify the similarity between two embeddings via multi-headed weighted cosine similarity, and thus it is a suitable choice in Eq. (7). To demonstrate the suitability and effectiveness of NTN, we extend our ablation study and ablate NTN from GraSP, and report the results. We find that NTN is beneficial in GraSP to improve performance. We have included the new ablation in Table 3 of the paper.
>
>
> **Table A.** Ablating NTN
> |                 | MSE ($\times 10^{-3}$) | $\rho$    | $\tau$    | P@10      | P@20      |
> |-----------------|------------------------|-----------|-----------|-----------|-----------|
> | GraSP           | **0.987**              | **0.930** | **0.829** | **0.806** | **0.863** |
> | GraSP (w/o NTN) | 1.473                  | 0.915     | 0.802     | 0.728     | 0.805     |
>
>
>
> > **Q1.** Numerous methods exist to enhance the expressiveness of GNNs. What motivated your decision to exclusively focus on positional encoding in your approach?
>
> **Response:** Positional encoding is well suited to the studied problem, because graph similarity is closely related to graph isomorphism testing (1-WL test) that determines whether two graphs are exactly the same. Therefore, we hope to enhance the expressive power of GNN to pass 1-WL, and as a result, improve the performance on graph similarity predictions, which motivates our decision to employ positional encoding.

---

> > ### Author Response · Authors · 2023-11-20
> > **Response to Reviewer dX6K (2/2)**
> >
> > > **Q2.** What is the rationale behind using RGGC as a backbone? Furthermore, can you explain how the gating mechanism contributes to the effectiveness of your task?
> >
> > **Response:** As explained in the paper, the RGGC layers can leverage a gating mechanism where gating units learn to control information flow through the network, which is beneficial to the task of graph similarity prediction. Furthermore, the gating mechanism is helpful in learning the importance of the messages passed from each node's neighbors, which is a feature beyond GIN or GCN. This is also beneficial for graph similarity prediction. The effectiveness of using RGGC is validated in Table 3 ablation study.
> >
> >
> > > **Q3.** On page 4 of your paper, you note that "both of the above pooling methods have some drawbacks." Can you offer more specific evidence or instances that highlight these limitations?
> >
> > **Response:** As explained in the paper, summation pooling treats all nodes equally, and attention in attention pooling runs the risk of overfitting on specific nodes. The specific evidence that highlights their limitations is provided below, as well as in Table 3 of the paper. Compared with GraSP that combines the advantages of both in a learnable way, the performance degrades when only using summation pooling, i.e., GraSP(w/o att), or only using attention pooling, i.e., GraSP(w/o sum).
> >
> > |                 | MSE ($\times 10^{-3}$)       | $\rho$    | $\tau$    | P@10      | P@20      |
> > |-----------------|-----------|-----------|-----------|-----------|-----------|
> > | GraSP           | **0.987** | **0.930** | **0.829** | **0.806** | **0.863** |
> > | GraSP (w/o att) | 1.089     | 0.923     | 0.817     | 0.789     | 0.844     |
> > | GraSP (w/o sum) | 1.077     | 0.922     | 0.817     | 0.779     | 0.847     |
> >
> > > **Q4.** Could you delve into the key differences between GED and MCS, explaining the importance of considering MCS when many related studies concentrate primarily on GED?
> >
> > **Response:** We highlight that our technical designs in GraSP are versatile to support more than one measure.  To apply GraSP to predict MCS and GED, the main difference is just the objective functions in Section 3.3, while the other designs in GraSP are the same.  GED and MCS have different definitions. MCS is the maximum common subgraph shared by two graphs, and it is suitable for the scenarios that require local matching of shared subgraph patterns, while GED is defined based on graph edit operations. Both MCS and GED are popular in quantifying graph similarities for applications in bioinformatics, chemistry, recommender systems, and social network analysis.

---

> > > ### Comment · Reviewer_dX6K · 2023-11-22
> > > **Acknowledgement of Rebuttal**
> > >
> > > Thank you for the response and conducting the additional experiments. After reviewing author's rebuttal and the comments from other reviewers, I'm not intend to elevate my rating.

---

> ### Author Response · Authors · 2023-11-22
> **Reminder for discussion.**
>
> Dear Reviewer dX6K,
>
> We thank your recognition of the excellent soundness of our work. Your insightful comments are solvable, and we have addressed them all and improved our paper. This is a reminder for discussion. Your feedback is important to us.
>
> Summary of responses:
> - We have conducted new experiments to ablate NTN (**Q5**), and clarified the purpose of NTN in our method (**W3**).
> - We have explained the efficiency of our method with experimental results (**W2, Q6**).
> - We have provided evidence on the design of our pooling technique (**Q3**).
> - We have improved the presentation of the paper (**W1**).
> - We have explained the motivation of our technical designs (**Q1, Q2**), and explained the difference between GED and MCS, and their importance (**Q4**).
>
> Best,
> Authors

---

### Official Review · Reviewer_NSFb · 2023-10-27

**Soundness:** 1 poor
**Presentation:** 2 fair
**Contribution:** 2 fair
**Rating:** 3
**Confidence:** 5

**Summary:**

This paper proposed GRASP, a method that leverages Graph Neural Networks to approximate Graph Distance/Similarity metrics, namely:
1. Graph Edit Distance (GED) and
2. Maximum Common Subgraph (MCS)

whose exact computations are NP-Hard. The authors enhanced node features using positional encoding and learned an embedding for graph using RGGC layers and multi-scale pooling. These embeddings are used to estimate GED/MCS. The authors demonstrated better efficacy and efficiency of their model compared to baselines.

**Strengths:**

1. The paper focuses on predicting graph similarity/distance metrics which is a very important problem. The organization of the paper is good and easy to follow.
2. The authors used positional encoding to enhance node features which also aided in passing the 1-WL test.
3. The ablation study is good which covered most of the design choices the authors made.

**Weaknesses:**

1. The authors presented MSE on predicted similarity scores (obtained by exponentiating a normalized version of GED) instead of predicted GED. Given the task is to predict GED, results should be reported on GED itself and not its transformation to a similarity metric that distorts the true errors. While some previous works such as SIMGNN have also followed the same methodology of reporting results on exponentiated similarity instead of true GED, there is no justification for this transformation.

2. The statistics of the data used for training, validation, and inference are not provided in the paper, which are important to understand the quality of results.

3. The MSE scores of baselines reported in the paper do not align with the existing literature. For example, GREED outperforms other baselines such as H2MN, SIMGNN in the literature but it is not reflected in this paper. What is the source of this discrepancy? Were all methods trained on error over GED or over the similarity score. The authors need to release the version of code they used benchmarking the baselines and the exact loss function so that reproducibility and any source of discrepancy can be properly analyzed.

4. Experiments are not extensive.
* The heat maps are provided for the AIDS700nef dataset where graphs are of small sizes. Heatmaps on datasets with larger graphs such as PTC give a better idea of the performance of Grasp and those need to be included.
* It is not clear how Grasp generalizes for unseen graph sizes. Given that generating ground truth for graphs with larger sizes is expensive, it is interesting to see how Grasp performs when training is done with smaller graphs and testing is done on larger unseen graphs. This aspect needs to be compared with other state-of-the-art baselines such as GREED and H2MN.

5. The estimated GED didn’t include costs for relabeling of edges. The authors didn’t mention how to extend this work to include edge substitution costs.

6. The novelty of GRASP is limited apart from using positional encoding. Grasp tackles the issue of cross-graph interactions, although it's important to recognize that Greed had already addressed this limitation before Grasp.

---------------------------
Overall, I am willing to revisit the rating, if the authors address concerns on reproducibility (release of baseline implementations used, train-test-validation stats, loss function clarifications), reported results include performance on true GED instead of transformed similarity that distorts performance due to exponentiation, and more detailed experiments in terms of generalizability to unseen, larger sizes, heatmaps, etc.

**Questions:**

1. It's unclear whether the reported MSE scores for GRASP are a result of the loss computed using the GED or similarity scores (obtained after normalization of GED). The code, specifically Line 170 in src/trainer.py, appears to calculate the loss using GED/Similarity score based on the command line parameters. A consistent methodology is required across all datasets and baselines. Hence, this needs to be clarified.

2. Are the baseline models trained to output GED scores or similarity scores (obtained after normalization of GED)? Models such as GREED are trained to output GED, training them to output similarity scores might affect the performance.

3. How is the Neural Tensor Network (NTN) affecting the performance of the method? Have you considered using only L2-Norm to predict GED, which will preserve the metric property as well? This analysis is not included in the ablation study.

4. What are the statistics of train-validation-test sets?

5. How does the accuracy vary with query size and GED on datasets with larger graphs?

6. What are the RMSE scores of GRASP and other baselines on GED?

---

> ### Author Response · Authors · 2023-11-20
> **Response to Reviewer NSFb and Look Forward to Your Reply (1/3)**
>
> We thank your insightful comments, and we have thoroughly addressed your comments via extensive experiments and detailed explanations.
> We look forward to your reply.
> 1. We have justified the GED training setting, and conducted new experiments to predict GED by all methods (**W1, Q2, Q6**).
> 2. We have clarified the data split ratio (**W2, Q4**), explained the source of discrepancy, and demonstrated the reproducibility of the experiments by experimental results and releasing all codes of baselines (**W3, Q1**).
> 3. We have provided the heatmaps of all datasets (**W4(a)**).
> 4. We have conducted new experiments to evaluate the generalizability of GraSP for unseen large graphs (**W4(b), Q5**).
> 5. As suggested, we highlighted our novelty (**W6**), and discussed how to extend the work to include edge relabeling (**W5**).
> 6. We have conducted a new ablation study on NTN in GraSP (**Q3**).
>
>
> > **W1.** The authors presented MSE on predicted similarity scores (obtained by exponentiating a normalized version of GED) instead of predicted GED. Given the task is to predict GED, results should be reported on GED itself and not its transformation to a similarity metric that distorts the true errors. While some previous works, such as SIMGNN have also followed the same methodology of reporting results on exponentiated similarity instead of true GED, there is no justification for this transformation.
> > **Q2.** Are the baseline models trained to output GED scores or similarity scores (obtained after normalization of GED)? Models such as GREED are trained to output GED, training them to output similarity scores might affect the performance.
> > **Q6.** What are the RMSE scores of GRASP and other baselines on GED?
>
> **Response:** Same as GREED, our method GraSP is designed to predict the original GED directly. However, note that the majority of existing methods, except GREED, are all trained to predict a normalized version of GED. Echoing your comment in Q2 on minimizing the impact on the performance of existing methods, we wish to make the least changes to existing methods and make the evaluation metric values comparable. Therefore, in the paper (i) we trained GraSP and GREED to output GED predictions directly, and then, as stated in Section 5.1, we normalize their predictions to calculate evaluation metrics; (ii) for the other baselines, we also followed their original design of using normalized GED for training and prediction. Note that RMSE is just the square root of MSE that is used in our paper. Upon your request in W1 and Q6, here we change the official codes of all baselines to predict GED directly, and report the RMSE results below, while including the complete results in Appendix A.7. Observe that GraSP consistently achieves better RMSE than existing methods for directly predicting GED on all datasets.
>
> **Table A.** The RMSE of all methods when predicting GED directly
>
> |     | AIDS700nef | IMDBMulti | LINUX | PTC |
> | --- | --- | --- | --- | --- |
> | GraSP | **0.808** | **7.924** | **0.137** | **4.826** |
> | GREED | *0.884* | *8.171* | 0.414 | *4.970* |
> | ERIC | 0.971 | 9.032 | 0.285 | 6.401 |
> | EGSC | 0.946 | 9.194 | *0.215* | 6.882 |
> | H2MN | 0.924 | 25.401 | 0.559 | 8.090 |
> | MGMN | 1.015 | 29.887 | 0.587 | 6.767 |
> | GMN | 0.947 | 13.027 | 0.369 | 7.551 |
> | GraphSim | 1.153 | 31.372 | 0.232 | 11.66 |
> | SimGNN | 1.008 | 18.041 | 0.518 | 8.364 |

---

> ### Author Response · Authors · 2023-11-20
> **Response to Reviewer NSFb (2/3)**
>
> > **W2 & Q4.** What are the statistics of train-validation-test sets?
>
> **Response:** We split training, validation, and testing data with a ratio of 6:2:2 for all datasets and all methods, by following the same setting as SimGNN and GraphSim.
>
>
>
> > **W3.** The MSE scores of baselines reported in the paper do not align with the existing literature. For example, GREED outperforms other baselines such as H2MN, SIMGNN in the literature but it is not reflected in this paper. What is the source of this discrepancy? Were all methods trained on error over GED or over the similarity score. The authors need to release the version of code they used benchmarking the baselines and the exact loss function so that reproducibility and any source of discrepancy can be properly analyzed.
> **Q1.** It's unclear whether the reported MSE scores for GRASP are a result of the loss computed using the GED or similarity scores (obtained after normalization of GED). The code, specifically Line 170 in src/trainer.py, appears to calculate the loss using GED/Similarity score based on the command line parameters.
>
>
> **Response:** The original papers of existing methods usually adopt *different* data split ratios. For example, GREED paper uses "100K query-target pairs for training and 10K pairs each for validation and test". In our paper, to be consistent across all methods, we split every dataset in the ratio of 6:2:2. A method can have different results with different data split ratios, which is a main source of discrepancy.
> In the original paper of GREED, three datasets AIDS, Linux, and IMDBMulti are used for GED predictions. In Table A of the response to **W1** above, the RMSE results of GREED on AIDS, Linux, IMDBMulti are on par with the original paper of GREED, which shows our reproducibility on baselines. Moreover, in **Table 1** of our paper, GREED outperforms H2MN by 10 out of 15 metrics  and SIMGNN by 12 out of 15 metrics on the three datasets, which again illustrates the reproducibility of our experiments. At Line 170 in src/trainer.py, we just provide users with the command-line options to train by either GED or similarity scores. As suggested, we have included the codes of all methods in the anonymous code repository.
>
> > **W4.(a)** Heatmaps on datasets with larger graphs such as PTC.
>
> **Response:** As suggested, we have included the heatmaps of PTC, LINUX, and IMDBMulti in Figures 9,10,11 of Appendix A.5 in the revised paper. The new heatmaps also show that our method GraSP achieves low error, indicated by a light color.
>
> > **W4**(b) It is not clear how Grasp generalizes for unseen graph sizes. Given that generating ground truth for graphs with larger sizes is expensive, it is interesting to see how Grasp performs when training is done with smaller graphs and testing is done on larger unseen graphs. This aspect needs to be compared with other state-of-the-art baselines such as GREED and H2MN.
> > **Q5.** How does the accuracy vary with query size and GED on datasets with larger graphs?
>
> **Response:** As suggested, we conducted new experiments by following the experimental setting in GREED to test the generalization ability of GraSP on large unseen graphs for GED predictions, compared to GREED and H2MN. Specifically, we get GraSP-25 and GraSP-50 by training GraSP on graphs with node sizes up to 25 and 50, respectively, and then test on graph pairs in which the number of nodes in the query graph is in the range [25, 50]. Table B shows the results of PTC data. Observe that (i) the performance of all methods degrades for large query sizes in [25,50], compared with the entire set in [0,50], (ii) when trained using smaller graphs from 50 to 25 size, all methods also degrade, (iii) our method GraSP-25 (resp. GraSP-50) keeps the best performance than baselines GREED and H2MN under all these generalization settings.
>
> **Table B.** Generalization on large unseen graphs with RMSE results to predict GED directly.
>
> |     | Query Size in [0,50] | Query Size in [25,50] |
> | --- | --- | --- |
> | GraSP-50 | **5.418** | **6.104** |
> | GREED-50 | 5.885 | 7.470 |
> | H2MN-50  | 7.041 | 9.024 |
> | GraSP-25 | **7.499** | **9.841** |
> | GREED-25 | 8.341 | 10.204 |
> | H2MN-25  | 8.965 | 10.956 |

---

> ### Author Response · Authors · 2023-11-20
> **Response to Reviewer NSFb (3/3)**
>
> > **W5.** The estimated GED didn’t include costs for relabeling of edges. The authors didn’t mention how to extend this work to include edge substitution costs.
>
> **Response:** As suggested, we have added the explanation on how to extend our method to include the cost of relabeling edges in Appendix A.8. Specifically, one possible way is that, instead of the gate $\eta_{i,j}$ in Eq. (3) of the paper, with reference to [1], we can modify Eq.(3) to adopt an edge gate $e^{(\ell)}_{i,j}$ :
>
> $h_i^{(\ell)} = h_i^{(\ell - 1)} + {ReLU}(W_1 h_i^{(\ell - 1)}  + {\underset{j \in \mathcal{N}(i)}{\sum}} {e_{i,j}}^{(\ell)}\odot W_2h_j^{(\ell - 1)})$, where $e_{i,j}^{(\ell)}$ is the edge gate and can be defined as:
> $e_{i,j}^{(\ell)} = \sigma(e_{i,j}^{(\ell - 1)} + ReLU(A h_i^{(\ell - 1)} + B h_j^{(\ell - 1)} + Ce_{i,j}^{(\ell - 1)}))$, where $A,B,C \in \mathbb{R} ^ {2d \times 2d}$ and $e^{(0)}_{i,j}$ represents the input edge features.
>
>
> [1] Vijay Prakash Dwivedi et al. Benchmarking Graph Neural Networks. Journal of Machine Learning Research, Volumn 24. 2023.
>
>
> > **W6.** The novelty of GRASP is limited apart from using positional encoding. Grasp tackles the issue of cross-graph interactions, although it's important to recognize that Greed had already addressed this limitation before Grasp.
>
> **Response:** We acknowledge that the issue of cross-graph interactions has been studied in the literature. Our novelty lies in the different technical designs to mitigate this issue, compared with existing methods. In the proposed GraSP, for the first time, positional encoding is introduced for the task of graph similarity prediction to enhance node features for higher expressiveness; GraSP further employs a graph neural network with gating and residual connections, and utilizes a multi-scale pooling technique to generate meaningful representation for accurate graph similarity predictions. Our theoretical analysis shows that GraSP can exceed 1-WL test. Notably, GraSP is versatile to accurately predict GED and MCS, and achieves superior performance across various datasets.
>
>
> > **Q3.** How is the Neural Tensor Network (NTN) affecting the performance of the method? Have you considered using only L2-Norm to predict GED, which will preserve the metric property as well? This analysis is not included in the ablation study.
>
> **Response:** As suggested, we ablate NTN in GraSP and report the results below. Observe that NTN helps GraSP to improve performance on all datasets, since NTN can be considered as a complement to the Euclidean distance over the graph embeddings in Eq. 6,7,8 of the paper. We have included the new ablation into Table 3 of the revised paper.
>
>
>
> **Table C.** Ablating NTN
> |                 | MSE ($\times 10^{-3}$) | $\rho$    | $\tau$    | P@10      | P@20      |
> |-----------------|------------------------|-----------|-----------|-----------|-----------|
> | GraSP           | **0.987**              | **0.930** | **0.829** | **0.806** | **0.863** |
> | GraSP (w/o NTN) | 1.473                  | 0.915     | 0.802     | 0.728     | 0.805     |

---

> > ### Comment · Reviewer_NSFb · 2023-11-22
> > **The empirical methodology remains problematic**
> >
> > I appreciate the additional experiments conducted by the reviewer. But my major concerns remain unaddressed.
> >
> > 1. The metric used to evaluate is not consistent with the problem formulation in Sec 2. The metric exponentiates the true distance and thus distorting the real error. In this version, errors between larger distances are largely ignored. As justification, the authors claim "the majority" of the works use this version. This line of argument is fallacious:
> >    * First, majority using a method does not make it the correct method. If the authors feel that the proposed metric should indeed be the metric of choice, it needs to be supported with proper arguments on what benefits exponentiation provides. Second, it is only some recent neural methods who have applied this metric. Approximating GED is a well-studied area and such exponentiation has not been used in many works [1][2][3][4]. Calling it majority is perhaps an exaggeration.
> >         -  [1] Huahai He and A. K. Singh, "Closure-Tree: An Index Structure for Graph Queries," 22nd International Conference on Data Engineering (ICDE'06), Atlanta, GA, USA, 2006, pp. 38-38, doi: 10.1109/ICDE.2006.37.
> >
> >         - [2] Zeng, Zhiping & Tung, Anthony & Wang, Jianyong & Feng, Jianhua & Zhou, Lizhu. (2009). Comparing Stars: On Approximating Graph Edit Distance.. PVLDB. 2. 25-36.
> >
> >        - [3] Jiyang Bai and Peixiang Zhao. 2021. TaGSim: type-aware graph similarity learning and computation. Proc. VLDB Endow. 15, 2 (October 2021), 335–347. https://doi.org/10.14778/3489496.3489513
> >
> >       - [4] Khoa D. Doan, Saurav Manchanda, Suchismita Mahapatra, and Chandan K. Reddy. 2021. Interpretable Graph Similarity Computation via Differentiable Optimal Alignment of Node Embeddings. In Proceedings of the 44th International ACM SIGIR Conference on Research and Development in Information Retrieval (SIGIR '21). Association for Computing Machinery, New York, NY, USA, 665–674. https://doi.org/10.1145/3404835.3462960
> >
> >    * Due to exponentiation, Table 2 misrepresents facts. See the massive discrepancy between Table 2 and Table 3 in [4]. For example, GraphSim which is one of the top performers as per Table 2 here, consistently produces poor performance when evaluated on true GED. While the authors incorporate some experiments on true GED errors in the revision, these are pushed to the appendix.
> >
> > 2. Why is GREED being trained on GED when the metric being measured is on the exponentiated version (for Table 2)?  It's an easy change in loss function. For a fair comparison, it demands that the loss function is aligned with the metric being used. That is not the case and hence the experiments paint a wrong picture.
> >
> > 3. I also find it surprising that while the authors report some generalizability results on graphs of unseen sizes in the openreview comments, they are not included in the revision? Why?
> >
> > Overall, the empirical methodology remains flawed. The metric being measured is not consistent with the problem formulation. Hence, Table 2 does not report the best performers on GED although it claims to do so. When computed on actual GED, the results change drastically (See [4]). Finally, the loss functions of the benchmarked algorithms, particularly GREED, which is the state of the art as per the literature, is not aligned to the metric although this is an easy change. No reasonable justification has been provided for this discrepancy.

---

> > > ### Author Response · Authors · 2023-11-23
> > > **Further responses and correction on inaccurate comments (1/2)**
> > >
> > > Dear reviewer NSFb,
> > >
> > > Please find our responses below. We also point out some inaccurate comments. Look forward to your reply.
> > >
> > > > **1(a)** The metric used to evaluate is not consistent with the problem formulation in Sec 2. The metric exponentiates the true distance and thus distorting the real error. In this version, errors between larger distances are largely ignored. As justification, the authors claim "the majority" of the works use this version. This line of argument is fallacious. First, majority using a method does not make it the correct method.
> > >
> > > **Response:** When saying "majority", it refers to the majority of *learning-based* graph similarity baselines, SimGNN, GraphSim, MGMN, H2MN, EGSC and ERIC, all of which use exponential normalization, rather than the vast literature from DB venues. If we make significant changes to all baselines to train via original GED, some other readers who may be the authors of these baselines will also complain, why not follow the majority setting of learning-based methods to normalize? Therefore, we respect the original design of all baselines to train, with minimized changes to their codes. Moreover, we have provided the results of original GED upon your request. We think this comment is unfair to us.
> > >
> > > > **1(b)** Due to exponentiation, Table 2 misrepresents facts. See the massive discrepancy between Table 2 and Table 3 in [4]. For example, GraphSim which is one of the top performers as per Table 2 here, consistently produces poor performance when evaluated on true GED. While the authors incorporate some experiments on true GED errors in the revision, these are pushed to the appendix.
> > >
> > > **Response:** We want to correct you that Table 2 in our paper is for **MCS** predictions, while Table 3 in [4] is for **GED** predictions, and of course, they are not comparable (we guess you are talking about Table 1, not Table 2, in our paper).
> > >
> > > For Table 1 in our paper, we kindly point out that **your observation that GraphSim is one of the top performers is not always true, and the performance of GraphSim is consistent with Table 3 in [4]**. In **Table 1** in our paper, among 10 methods:
> > > - On AIDS700nef, except for the MSE metric, GraphSim is outperformed by **6 methods**, including MGMN, H2MN, EGSC, ERIC, GREED, and our GraSP, on $\rho$, $\tau$, $P@10$, and $P@20$.
> > > - On IMDBMulti, GraphSim is outperformed by **5 methods**, including H2MN, EGSC, ERIC, GREED, and GraSP, on all metrics.
> > > - On LINUX and PTC, GraphSim performs well, which is **consistent** with Table 3 in [4] where GraphSim is also top.
> > >
> > > [4] Interpretable Graph Similarity Computation via Differentiable Optimal Alignment of Node Embeddings.

---

> ### Author Response · Authors · 2023-11-22
> **Reminder for discussion.**
>
> Dear Reviewer NSFb,
>
> We appreciate your willingness to revisit the rating. We have put considerable effort into addressing all your constructive comments, including experiments on reproducibility (release of baseline implementations, train-test-validation stats, loss function clarifications), performance on true GED, and more experiments in terms of generalizability to unseen, larger sizes, heatmaps, etc. We summarize the responses below. Please have a look and discuss. Thanks.
>
> - As suggested, we have clarified the GED training setting, and conducted new experiments to predict GED by all methods (**W1, Q2, Q6**).
> - We have provided the data split ratio (**W2, Q4**), explained the source of discrepancy, demonstrated the reproducibility of experimental results, and released all codes of baselines (**W3, Q1**).
> - We have provided the heatmaps of all datasets (**W4(a)**).
> - We have conducted new experiments to evaluate the generalizability of GraSP for unseen large graphs (**W4(b), Q5**).
> - As suggested, we highlighted our novelty (**W6**), and discussed extending the work to include edge relabeling (**W5**).
> - We have conducted a new ablation study on NTN in GraSP (**Q3**).
>
> Best,
> Authors

---

> ### Author Response · Authors · 2023-11-23
> **Further responses and correction on inaccurate comments (2/2)**
>
> > **2.** Why is GREED being trained on GED when the metric being measured is on the exponentiated version (for Table 2)? It's an easy change in loss function. For a fair comparison, it demands that the loss function is aligned with the metric being used. That is not the case and hence the experiments paint a wrong picture.
>
> **Response:**  Again, we guess it should be Table 1 in our paper, instead of Table 2. (i) As explained, to minimize the changes to baselines, GREED was trained with its *original* loss function on GED, while the other baselines also followed their *original* design to train over exponentially normalized GEDs. (ii) Upon your previous comment, we have already provided the results of GREED and all methods trained on original GED, where GREED is the top baseline, as shown in Table A above and Table 6 in the revised paper.  (iii) Here, we further revise the loss function of GREED in the code to cope with similarity scores. We provide the results of GREED trained on similarity scores in Table D(1) below. We also copy the results of GREED trained on original GEDs from Table 1 in the paper to Table D(2) for comparison. Note that the MSE in Table D(2) is calculated by exponentiating the predicted GEDs as we discussed on Section 5.1 in the paper. We observe that all metrics, including the MSE, are comparable for these two versions of GREED. Therefore, we conclude that our experiments are fair and reproducible.
>
> **Table D(1).** GREED, trained on similarity scores.
> |            | MSE ($\times 10 ^{-3}$) | $\rho$ | $\tau$ | P@10  | P@20  |
> |------------|-------------------------|--------|--------|-------|-------|
> | AIDS700nef | 1.773                   | 0.895  | 0.751  | 0.579 | 0.651 |
> | IMDBMulti  | 0.741                   | 0.911  | 0.828  | 0.852 | 0.843 |
> | LINUX      | 0.867                   | 0.974  | 0.904  | 0.959 | 0.951 |
> | PTC        | 2.354                   | 0.932  | 0.794  | 0.467 | 0.574 |
>
> **Table D(2).** GREED, trained on original GEDs (from Table 1 of the paper).
> |            | MSE ($\times 10 ^{-3}$) | $\rho$ | $\tau$ | P@10  | P@20  |
> |------------|-------------------------|--------|--------|-------|-------|
> | AIDS700nef | 1.432                   | 0.913  | 0.796  | 0.710 | 0.780 |
> | IMDBMulti  | 1.174                   | 0.930  | 0.865  | 0.859 | 0.858 |
> | LINUX      | 0.926                   | 0.966  | 0.905  | 0.978 | 0.975 |
> | PTC        | 2.442                   | 0.889  | 0.765  | 0.424 | 0.517 |
>
>
> > **3.** I also find it surprising that while the authors report some generalizability results on graphs of unseen sizes in the openreview comments, they are not included in the revision? Why?
>
> **Response:** It is included in appendix A.9 of the revised paper now.

---

### Official Review · Reviewer_Ytzi · 2023-10-31

**Soundness:** 3 good
**Presentation:** 3 good
**Contribution:** 2 fair
**Rating:** 6
**Confidence:** 4

**Summary:**

The paper proposes a new method for approximating graph similarity scores, leveraging random walk method for position encoding, RWPE, and multi-scale pooling. The overall method GRASP shows superior performance compared to exiting methods over four datasets and two graph similarity/distance metrics.

**Strengths:**

1. The method is well-motivated with the observation that cross-graph node-level interaction is costly, and the proposed method is relatively simple with great inference time reduction.
2. The paper is well written and easy to follow.

**Weaknesses:**

1. It would be interesting to know what would be the alternative to the proposed combination of summation and attention pooling, e.g. what if a simple non-learning combination is used, i.e. $\alpha * z_{sum} + (1 - \alpha) * z_{att}$ where $\alpha$ is a hyperaprameter scalar. This will further highlight the importance of learnable combination of the two pooling methods.

2. The overall novelty is limited, given the adoption of random walk positional encoding and combination of existing graph pooling are not firstly proposed in this paper.

**Questions:**

1. Why the time complexity for computing the random walk positional encoding can be omitted? For each new graph pair, the inference time should include such positional encoding, since at inference time, the graph pair can be new and thus unseen by the model. Unless the authors assume a database-like setting, such overhead cannot be omitted.

---

> ### Author Response · Authors · 2023-11-20
> **Response to Reviewer Ytzi and Look Forward to Your Reply**
>
> We thank your insightful comments, and we have thoroughly addressed them as follows. We look forward to your reply.
>
> > **W1.** It would be interesting to know what would be the alternative to the proposed combination of summation and attention pooling, e.g. what if a simple non-learning combination is used, i.e. $\alpha * z_{sum} + (1-\alpha)* z_{att}$ where $\alpha$ is a hyperparameter scalar. This will further highlight the importance of learnable combination of the two pooling methods.
>
> **Response:** As suggested, we vary a non-learning scalar value $\alpha$ from 0 to 1, with MSE results on GED prediction in Table A below, where the last row shows GraSP with the proposed learnable combination. Observe that (i) the proposed learnable combination in GraSP achieves the best performance on all datasets; (ii) the non-learning combination needs different $\alpha$ scalar values on different datasets, which is tedious to tune. These observations highlight the importance of the proposed learnable combination of the poolings.
>
> **Table A.** MSE ($\times 10^{-3}$) results on learnable combination v.s. non-learning combination.
> | $\alpha$    | AIDS700nef | IMDBMulti | LINUX     | PTC       |
> |-------------|------------|-----------|-----------|-----------|
> | 0 (w/o sum) | 1.540      | 1.621     | 0.156     | 1.831     |
> | 0.2         | *1.433*      | 1.020     | 0.136     | 1.847     |
> | 0.4         | 1.590      | 1.478     | *0.134*     | *1.784*     |
> | 0.6         | 1.594      | 1.572     | 0.142     | 1.812     |
> | 0.8         | 1.505      | *0.986*     | 0.153     | 1.811     |
> | 1 (w/o att) | 1.556      | 2.272     | 0.156     | 1.813     |
> | learnable   | **0.987**  | **0.789** | **0.075** | **1.641** |
>
>
> > **W2.** The overall novelty is limited, given the adoption of random walk positional encoding and combination of existing graph pooling are not firstly proposed in this paper.
>
> **Response:** Compared with existing methods for graph similarity predictions, our novelty lies in the different technical designs to achieve superior performance for predicting GED and MCS. To our knowledge, we are the first to introduce positional encoding to the graph similarity predictions to node features for higher expressiveness. Furthermore, GraSP employs a graph neural network with gating and residual connections, and utilizes a multi-scale pooling technique to generate meaningful representation for accurate graph similarity predictions. Our theoretical analysis shows that GraSP can exceed 1-WL test. Notably, GraSP achieves superior performance across various datasets.
>
>
>
> > **Q1.** Why the time complexity for computing the random walk positional encoding can be omitted? For each new graph pair, the inference time should include such positional encoding, since at inference time, the graph pair can be new and thus unseen by the model. Unless the authors assume a database-like setting, such overhead cannot be omitted.
>
> **Response:** Note that the positional encoding is applied per graph, instead of per graph pair. During online inference, if a graph appears in multiple graph pairs, the positional encoding of the graph is computed only once, but used many times. Therefore, given a dataset of graphs, we can pre-compute and store the positional encoding of every graph. Then, during online inference, given a graph pair, we only need to retrieve the encodings of the two graphs in the pair and reuse them as the input of GraSP.  This explains why the time complexity of positional encoding can be omitted for inference time.

---

> ### Author Response · Authors · 2023-11-22
> **Reminder for discussion.**
>
> Dear Reviewer Ytzi,
>
> We are grateful for your appreciation on our work. This is a reminder for discussion. We have addressed all your constructive comments in the responses. Thanks.
>
> Best,
> Authors

---

### Meta-Review · Area_Chair_q17U · 2023-12-09

**Metareview:**

This paper proposed GRASP, a method that leverages Graph Neural Networks to approximate Graph Distance/Similarity metrics whose exact computations are NP-Hard. The authors enhanced node features using positional encoding and learned an embedding for graph using RGGC layers and multi-scale pooling. These embeddings are used to estimate GED/MCS. The authors demonstrated better efficacy and efficiency of their model compared to baselines. The paper has the following limitations: 1. Presentation and Clarity Issues: Reviewers highlighted the lack of clarity, particularly in Section 3, and the overall cohesiveness of the paper, making it challenging for readers to grasp the proposed methodology. 2. Limited Novelty and Inconsistency in Methodology: Concerns are raised about the novelty of GRASP's approach and inconsistencies in its methodological framework, especially regarding the computation of similarity scores and the use of exponentiation. 3. Inadequate Empirical Support for Efficiency Claims: While GRASP claims to mitigate inefficiencies in cross-graph interactions, reviewers noted the lack of conclusive empirical evidence to support this. Reviewer NSFb, in particular, emphasizes the need for more thorough evaluation, including aligning the metric with the problem formulation and redoing experiments with metrics purely on GED. In summary, while GRASP demonstrates potential, substantial modifications and improvements are required for a more comprehensive and accurate presentation of its capabilities.

**Justification For Why Not Higher Score:**

Inconsistency in experiment design and metrics.

**Justification For Why Not Lower Score:**

N/A

---

### Decision · Program_Chairs · 2024-01-16

Reject